# Mesh Field Theory: Port–Hamiltonian Formulation of Mesh-Based Physics

**Satoshi Noguchi** [1] [2]   **Yoshinobu Kawahara** [3] [2]

## Abstract

We present *Mesh Field Theory (MeshFT)* and its neural realization, MeshFT-Net: a structure-preserving framework for mesh-based continuum physics that cleanly separates the physics' topological structure from its metric structure. Imposing minimal physical principles (locality, permutation equivariance, orientation covariance, and energy balance/dissipation inequality), we prove a reduction theorem for mesh-based physics. Under these conditions, the physical dynamics admit a local factorization into a port–Hamiltonian form: the conservative interconnection is fixed uniquely by mesh topology, whereas metric effects enter only through constitutive relations and dissipation. This reduction clarifies what must be fixed and what should be learned, directly informing MeshFT-Net's design. Across evaluations on analytic and realistic datasets, physics-consistency tests, and out-of-distribution validation, MeshFT-Net achieves near-zero energy drift and strong physical fidelity—correct dispersion and momentum conservation—along with robust extrapolation and high data efficiency. By eliminating non-physical degrees of freedom and learning only metric-dependent structure, MeshFT provides a principled inductive bias for stable, faithful, and data-efficient physical simulation. The code is available at https://github.com/noguchisatoshi/MeshFT-Net.

## 1. Introduction

Machine learning is increasingly used to accelerate continuum simulations, from data-driven surrogates and weak-

[1]Center for Mathematical Science and Advanced Technology, JAMSTEC [2]Center for Advanced Intelligence Project, RIKEN [3]Graduate School of Information Science and Technology, The University of Osaka. Correspondence to: Satoshi Noguchi <satoshin@jamstec.go.jp>, Yoshinobu Kawahara <kawahara@ist.osaka-u.ac.jp>.

*Proceedings of the 43$^{rd}$ International Conference on Machine Learning*, Seoul, South Korea. PMLR 306, 2026. Copyright 2026 by the author(s).

form training to operator learning that transfers across meshes and parameters (Lu et al., 2021; Kovachki et al., 2023; E & Yu, 2018; Li et al., 2021; Gupta et al., 2021; Raissi et al., 2019; Sirignano & Spiliopoulos, 2018; Ummenhofer et al., 2020; Cao, 2021; Battaglia et al., 2018). Another line of work represents a discretized domain as a graph built from a mesh and learns time evolution by message passing, yielding strong results in analysis of fluid and deformable solids while remaining flexible in resolution and topology (Pfaff et al., 2021; Sanchez-Gonzalez et al., 2020).

However, a key structural point is often left implicit. In exterior calculus on a manifold (Flanders, 1963), the exterior derivative $d$ is *topological* (metric-independent) while geometry and material properties appear only through metric-dependent operators such as the Hodge star $\star$. In many existing learned mesh simulators, these roles are conflated, letting non-physical effects contaminate predicted fields and produce spurious modes and instabilities.

Discrete exterior calculus (DEC) (Hirani, 2003; Desbrun et al., 2005; 2006) argues that mesh topology provides the algebraic backbone for differential operators describing mesh-based physics. Also, metric-dependent part is clearly divided from such topological structures. Classical structure-preserving numerical schemes (e.g., finite-difference time-domain (FDTD)) exploits the same geometric structure and, as a result, yield stable and conservative updates (Yee, 1966; Taflove & Hagness, 2005; Noguchi et al., 2020; Bossavit, 1998). These observations suggest a clean division of roles in mesh-based physics simulators: hard-code topological structures based on the algebraic backbone given by fixed mesh topology, and learn only metric-dependent structures.

We develop this idea as *Mesh Field Theory (MeshFT)* and its neural realization MeshFT-Net. First, we formalize four physical requirements, locality, permutation equivariance, orientation covariance, and an energy balance/passivity inequality. Then, under these conditions, we prove that physical dynamics on a mesh follow a local port–Hamiltonian reduction in which the conservative interconnection is fixed by mesh topology, while metric effects enter only through constitutive and dissipative operators. Importantly, this local port–Hamiltonian structure is *deduced* from the given principles rather than assumed as a prescribed global template. Guided by the theorem, MeshFT-Net can implement the

principled inductive bias by fixing the topology-wired conservative coupling and learning only metric and dissipation factors, without assuming a known partial differential equation (PDE) form or a predefined global structure template.

Across evaluation on analytic datasets and acoustic-scattering data from *The Well* (Ohana et al., 2024; Mandli et al., 2016), together with physics-consistency tests and out-of-distribution (OOD) validation, MeshFT-Net achieves near-zero energy drift while preserving correct dispersion and momentum behavior, and it improves generalization and data efficiency relative to baselines. These results support that the principle-driven approach, which induces an explicit topology–metric separation, provides an inductive bias for stable and faithful mesh-based simulation.

Our contributions can be summarized as follows: **Reduction theorem.** We formalize four physical requirements and prove a principle-driven local reduction theorem. Without assuming a known PDE form or committing to a prescribed global dynamics template, the dynamics reduce locally to a port–Hamiltonian form in which the conservative interconnection has topology-fixed signed-incidence sparsity, while only metric-dependent constitutive and dissipative effects are learnable. **Neural architecture.** Based on the theorem, we design MeshFT-Net to fix the topology-wired incidence structure and learn only metric-dependent constitutive/dissipative operators. **Empirical results.** Across evaluation on acoustic-scattering data from *The Well*, physics-consistency tests, and OOD validation, MeshFT-Net shows near-zero energy drift with strong physical fidelity, robust extrapolation performance, and higher data efficiency.

## 2. Related Work

By comparison with related work, we position MeshFT as a principle-derived hypothesis class for mesh-based physics. Starting from physical principles, MeshFT deduces a Jacobian-level local reduction in which conservative coupling is constrained to a signed-incidence wiring determined by mesh topology, while learning is confined to local metric and dissipation operators. This differs from approaches that begin by postulating a global Hamiltonian or port–Hamiltonian template and then learning the remaining components within that form. Starting from principles avoids committing to a single global template, which improves robustness to model mismatch.

**MeshGraphNets (MGN).** MGN learn mesh-based dynamics via permutation-equivariant message passing on mesh graphs (Pfaff et al., 2021; Battaglia et al., 2018). From our viewpoint, MeshGraphNets can be seen as relaxations of MeshFT: by not enforcing the orientation and energy constraints, they can represent a broader class of interactions, which offers flexibility but does not guarantee a part of physical requirements and can admit spurious modes (Fig. 1).

**Structure-Preserving Learning of Dynamics.** Hamiltonian and Lagrangian neural networks learn an energy and induce a structured field, e.g., $(\partial_p H_\theta, -\partial_q H_\theta)$ in canonical coordinates (Greydanus et al., 2019; Cranmer et al., 2020; David & Méhats, 2023; Eidnes & Lye, 2024). In addition, some approaches adopt a port–Hamiltonian and learn ingredients such as a Hamiltonian and dissipation operators from data (Desai et al., 2021; Course et al., 2020; Neary & Topcu, 2023). Thermodynamically structured models such as metriplectic or GENERIC typically fit operators within a prescribed global template (Zhang et al., 2022; Hernández et al., 2021; Lee et al., 2021; Hernández et al., 2026; He et al., 2026). Symplectic neural ODE methods combine learned dynamics with symplectic integration (Zhong et al., 2020). These approaches are effective when an appropriate template is available. MeshFT instead deduces a local port–Hamiltonian structure from fundamental physical principles, which constrains conservative coupling by topology and leaves only local metric and dissipation factors to learn.

**Neural Constitutive Laws.** Constitutive-learning approaches often assume a known continuum PDE and learn unknown parameter fields (Ma et al., 2023). MeshFT does not assume an explicit continuum PDE or PDE-residual supervision. It fixes the topology-wired coupling and learns metric-dependent operators, which can be interpreted as learning constitutive relations in some cases.

**DEC and Data-Driven Exterior Calculus.** DEC formalizes structure-preserving discretizations in which cochains represent fields on $k$-cells and the coboundary operator $D_k$ satisfies $D_{k+1}D_k = 0$ (Hirani, 2003; Desbrun et al., 2005; Arnold et al., 2006). Metric-dependent effects enter through discrete Hodge operators, defining inner products, energies, and constitutive relations. Data-driven exterior calculus similarly fixes incidence operators and learns metric-dependent operators such as Hodge stars (Trask et al., 2022), typically within a discretized operator-learning setting. MeshFT is aligned with this topology–metric split, but starts from physical principles and uses the resulting Jacobian-level reduction to constrain admissible dynamics.

**Exterior-Calculus Related Networks.** Several network architectures incorporate exterior-calculus or bracket-inspired structure, for example GRAND (Chamberlain et al., 2021) and bracket-based GNNs (Gruber et al., 2023). Related work also uses differential-form ideas to construct constrained vector fields, including neural conservation laws (Richter-Powell et al., 2022) and learning symplectic forms (Chen et al., 2021). These provide geometric inductive biases at the layer or vector-field level. MeshFT focuses on mesh-based continuum dynamics where topology-fixed incidence constraints and local metric structure arise jointly from the physical principles through the reduction theorem.

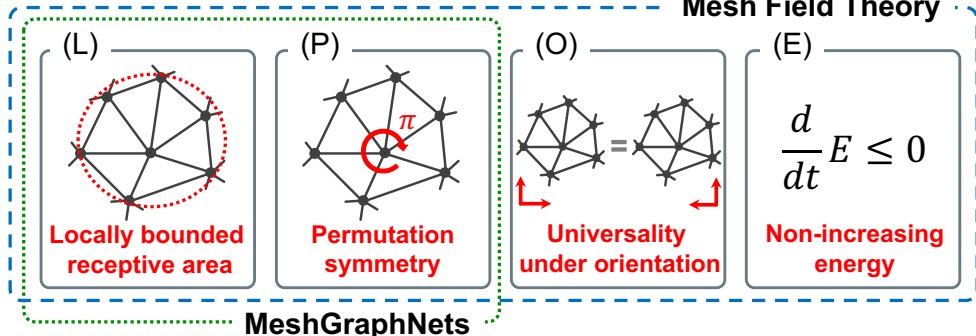

*Figure 1.* Core concept of this study—comparison of MeshFT and MGN by underlying physical assumptions: (L) locality, (P) Permutation equivariance, (O) Orientation covariance, (E) Non-increasing energy. MGN attains (L) and (P) by architecture design, whereas MeshFT additionally enforces (O) and (E), yielding a clear modeling guideline: *fix topology* (incidence-based interconnection) and *learn metric-dependent structures*, which directly leads to MeshFT-Net.

*Table 1.* Notation.

| Symbol | Definition |
| --- | --- |
| $n_k$ | Number of oriented $k$-cells. |
| $C^k$ | Space of $k$-cochains, $C^k \simeq \mathbb{R}^{n_k}$. Examples: $C^0$ nodes, $C^1$ oriented edges, $C^2$ oriented faces. |
| $z$ | Stacked dynamical state: $z \in C^k \oplus C^{k+1}$. |
| $D_k$ | Coboundary / signed incidence matrix: $D_k : C^k \to C^{k+1}$, $D_k \in \mathbb{R}^{n_{k+1} \times n_k}$. |
| $D_k^\top$ | Ordinary matrix transpose of $D_k$: $D_k^\top : C^{k+1} \to C^k$, $D_k^\top \in \mathbb{R}^{n_k \times n_{k+1}}$. |
| $H$ | Scalar storage energy: $H : C^k \oplus C^{k+1} \to \mathbb{R}_{\geq 0}$. |

## 3. Local Reduction Theorem

In this section, we establish a port–Hamiltonian formulation for mesh-based physics. We show that the dynamics of MGN reduce to a port–Hamiltonian form at the differential level under a minimal set of physical requirements, with several of these requirements already implicitly satisfied by vanilla MGN. Fig. 1 compares MeshFT and MGN in terms of their underlying assumptions, clarifying the paper's central idea. In particular, enforcing the orientation covariance and the energy balance eliminates non-physical degrees of freedom and reveals a clean separation between the topology-driven interconnection and the metric-dependent energetic part of the dynamics, thereby clarifying which must remain fixed and which components should be learned.

**Notation.** Let $\mathcal{K}$ be a finite oriented cell complex of dimension $d$. For each $k$, let $C^k \cong \mathbb{R}^{n_k}$ denote the space of real-valued $k$-cochains (one scalar per oriented $k$-cell), and let $D_k : C^k \to C^{k+1}$ be the signed incidence (coboundary) operator, with $D_{k+1} D_k = 0$ (Hirani, 2003; Desbrun et al., 2005). We stack the cochain degrees that interact via incidence (e.g., $k$ and $k+1$) as $z := (z_k, z_{k+1}) \in C^k \oplus C^{k+1}$, and consider an autonomous update $\dot{z} = F(z)$. For example, a one-layer update in MGN often takes $z = (z_0, z_1)$ with node features $z_0 \in C^0$ and edge features $z_1 \in C^1$ on *undi-*

*rected* edges. Such edge features are not DEC 1-cochains, and the coupling is implemented by permutation-invariant message passing rather than by the signed incidence $D_0$. Generally, $z$ need not be limited to nodes and edges, it may include face features ($C^2$) or cell-centered features ($C^d$). In addition, vector-valued features are handled by replacing $C^k$ with $C^k \otimes \mathbb{R}^{r_k}$. Also, non-physical features (e.g., node/edge types) are excluded from the state $z$ here. They may only enter as fixed parameters of learnable maps, and are not treated as dynamical variables.

Once orientations and an ordering of cells are fixed, we use the same symbol $D_k$ for the coboundary operator and for its sparse signed-incidence matrix. The symbol $D_k^\top$ denotes the transpose. For instance, $D_0$ maps node values to oriented edge differences, while $D_1$ maps oriented edge values to oriented face circulations. Table 1 summarizes the main object and a concrete example is given in Appendix A.

We equip the state $z = (z_k, z_{k+1}) \in C^k \oplus C^{k+1}$ with a storage function $H : C^k \oplus C^{k+1} \to \mathbb{R}_{\geq 0}$, strictly convex, and block-separable by degree, i.e., $H(z) = H_k(z_k) + H_{k+1}(z_{k+1})$. It is time-invariant but may encode spatial/material heterogeneity. Define the co-energy (conjugate) variable $e := \nabla H(z) \in C^k \oplus C^{k+1}$, where the gradient is taken with respect to the canonical Euclidean pairing, so the instantaneous power delivered to the state is $\langle e, \dot{z} \rangle = e^\top \dot{z}$. $G(z) := \nabla^2 H(z) \succ 0$ is block-diagonal across degrees and acts as a state-dependent metric. We write the associated mass map as $M(z) := G(z)^{-1} \succ 0$. The linear–quadratic case is recovered when $G(z) \equiv M^{-1}$ is constant, i.e., $H(z) = \frac{1}{2} z^\top M^{-1} z$ and $e = M^{-1} z$.

### 3.1. What MGN Guarantees by Design, and What Physics Still Requires

In this section, we identify physical requirements that MGN implicitly respects and the essential ones absent from the standard formulation despite their importance in faithful

physical modeling. In this section, we focus on the spatial structure and do not explicitly enforce additional temporal priors (e.g., causality), which are addressed by the causal time-stepping scheme.

**MGN Built-in Locality and Symmetry.** We note two inductive biases that MGN inherently satisfies—local interactions and permutation equivariance (label symmetry):

**Assumption 3.1** (MGN built-ins: Locality and Symmetry). **(L) Locality.** The output on a $k$-cell may depend only on inputs within $L$ hops in the graph of $\mathcal{K}$. For $L = 1$, a $k$-cell receives only from itself, its own faces $(k-1)$-cell, and its own cofaces $(k+1)$-cell. **(P) Permutation equivariance.** For any permutation $\pi$ that preserves the partition of indices by degree/type, the predictor satisfies $f(\pi \cdot x) = \pi \cdot f(x)$.

Informally, MGN realizes (L) and (P) by gathering messages (1-cochain) from 1-hop neighbor nodes (0-cell) with an order-independent reducer (e.g., sum/mean/max), applying a shared message/update kernel, and aggregating back to 1-hop neighbor nodes. This realizes permutation equivariance and locality, which happens to match common physical desiderata—local and uniform governing laws. Intuitive visualization is shown in Fig. 1. However, only these assumptions do not guarantee geometric/orientation correctness or energetic consistency. With these in place, we formalize the two additional physical requirements not enforced by MGN—orientation covariance and energy balance.

**Physical Requirements Beyond MGN.** The following two physical requirements, typically not enforced by standard MGN, will be added:

**Assumption 3.2** (Orientation & Energy). **(O) Orientation covariance.** Changing the sign convention of oriented entities (e.g., flipping edge/face directions) should only flip the signs of the corresponding oriented variables and scalar quantities such as energy and power must be unchanged. The formal flip-operator statement is given in Appendix B. **(E) Energy balance and passivity.** The dynamics split into a conservative part that never does net work and a dissipative part that never injects energy. Consequently, in the absence of sources the stored energy cannot increase over time.

Formally, we assume an energy balance with a conservative–dissipative split $F = F_{\text{con}} + F_{\text{diss}}$ and $e = \nabla H(z)$. *Pointwise* power satisfies, for all $z$, $e^\top F_{\text{con}}(z) = 0$ and $e^\top F_{\text{diss}}(z) \leq 0$, so $\dot{H} = e^\top \dot{z} = e^\top F(z) \leq 0$. We also impose the *incremental* (two-point) form in energy variables: for $z_1, z_2$ with $e_i = \nabla H(z_i)$, $(e_1 - e_2)^\top (F_{\text{con}}(z_1) - F_{\text{con}}(z_2)) = 0$, $(e_1 - e_2)^\top (F_{\text{diss}}(z_1) - F_{\text{diss}}(z_2)) \leq 0$. Thus the conservative part does no net work between states, while the dissipative part is *monotone* in $e$. If $F$ is differentiable, these incremental conditions are equivalent to $\text{Sym}\left(\frac{\partial F_{\text{con}}}{\partial e}\right) = 0$ and $\text{Sym}\left(\frac{\partial F_{\text{diss}}}{\partial e}\right) \preceq 0$ which yields the skew/dissipative split used in our reduction.

**Importance of Orientation Covariance** Orientation is a sign gauge: flipping the orientation of $k$-cells (edge arrows, face normals) changes coordinates but not physics. Let $\rho = \text{diag}(\rho_0, \ldots, \rho_d)$ with $\rho_k \in \{\pm I\}$ act on all oriented variables on $k$-cells. Scalars are gauge-invariant: $H(\rho z) = H(z)$ and $e(\rho z)^\top F(\rho z) = e(z)^\top F(z)$. Fluxes carried by oriented $k$-cells are gauge-covariant: $q_k \mapsto \rho_k q_k$. The signed incidence transforms as $D_k \rho_k = -D_k$, $\rho_{k+1} D_k = -D_k$, and $\rho_{k+1} D_k \rho_k = D_k$, corresponding to flipping degree $k$ only, degree $k+1$ only, or both simultaneously. Assumption 3.2 (O) ensures sign-gauge equivariance: the physical laws retain their form and scalar pairings (e.g., $e^\top z$) remain unchanged under orientation flips. In other words, this guarantees that the physical system is universal regardless of the choice of mesh orientation shown in Fig. 1.

### 3.2. Local Reduction to Port–Hamiltonian Dynamics

We show that mesh-based dynamics satisfying the MGN built-in principles (locality and permutation equivariance) together with the physical principles introduced above (orientation covariance and energy balance/passivity) admit a *local* reduction to port–Hamiltonian representation. We now state the reduction theorem. A complete proof of Theorem 3.3 appears in Appendix B.

**Theorem 3.3** (Local reduction to port–Hamiltonian dynamics). *Let $F : C^k \oplus C^{k+1} \to C^k \oplus C^{k+1}$ define the continuous-time dynamics $\dot{z} = F(z)$. Assume (L), (P), (O), and (E). At any point where the Jacobian exists, there is a local energy reparameterization under which*

$$\frac{\partial F}{\partial z}(z) = \big(J(z) - R(z)\big) G(z),$$

*where $J(z)^\top = -J(z)$, $R(z) \succeq 0$, and $G(z) \succ 0$. Moreover, up to cochain ordering and an orientation gauge, the off-diagonal blocks of $J(z)$ have signed-incidence wiring: for each adjacent pair $(k, k+1)$, $J_{k,k+1}(z) = -D_k^\top C_k(z)$, $J_{k+1,k}(z) = C_k(z) D_k$, with $C_k(z) = \text{diag}(c_k(z)) \succ 0$, where $c_k$ is local, permutation-equivariant, and orientation-even.*

**Remark.** Theorem 3.3 is an equation-agnostic local statement at the Jacobian level. We neither assume nor claim that the full dynamics admits a single global port–Hamiltonian representation. Rather, if $F$ is locally differentiable and satisfies (L/P/O/E), then the Jacobian admits the factorization above at points where the Jacobian exists. If $F$ is only piecewise smooth, the reduction holds on the smooth regions.

**Corollary 3.4** (State-independent interconnection). *Assume the incidence gains are* state-independent, *namely $C_k(z) \equiv C_k$ for all $z$ (they may still depend on fixed geometry or material). Then one may take a constant skew interconnection $J^\top = -J$ with $J_{k,k+1} = -D_k^\top C_k$ and $J_{k+1,k} = C_k D_k$, so that $\frac{\partial F}{\partial z}(z) = (J - R(z))G(z)$ holds*

*with the same $G(z)$ and $R(z)$ as in Theorem 3.3. Moreover, with the constant rescaling $S = \operatorname{diag}(I, C_k)$ and $\tilde{F}(\tilde{z}) = S^{-1} F(S\tilde{z})$, the Jacobian satisfies*

$$\frac{\partial \tilde{F}}{\partial \tilde{z}}(\tilde{z}) = \left(\tilde{J} - \tilde{R}(\tilde{z})\right) \tilde{G}(\tilde{z}),$$

$\tilde{J}_{k,k+1} = -D_k^\top$, $\tilde{J}_{k+1,k} = D_k$, *where* $\tilde{G}(\tilde{z}) = S^\top G(S\tilde{z})S$ *and* $\tilde{R}(\tilde{z}) = S^{-1} R(S\tilde{z}) S^{-\top}$.

The novelty is not merely to posit a port–Hamiltonian form, but to characterize what is fixed and what is learnable, and to give sufficient conditions for this separation. Under (L), (P), (O), and (E), the conservative coupling is constrained to a signed-incidence wiring dictated by the mesh topology, while phenomenon-dependent effects are carried by the metric-side factors and dissipation through $G(z)$ and $R(z)$. Accordingly, learning reduces to parameterizing these local operators and the admissible incidence-modulating coefficients $C_k(z)$ within the fixed sparsity and sign structure of the interconnection. In particular, when the incidence gains are state-independent one may choose a constant interconnection as in Corollary 3.4.

In this paper, we henceforth adopt the state-independent setting of Corollary 3.4 and work in the incidence-normalized coordinates, so the conservative coupling is represented by the pure signed-incidence blocks $\tilde{J}_{k,k+1} = -D_k^\top$ and $\tilde{J}_{k+1,k} = D_k$. For notational simplicity, we identify $\tilde{J}$ with $J$ in what follows. This assumption still covers many standard settings including linear waves, linear elastodynamics, and linear electromagnetics on fixed media, with heterogeneity entering through the metric and dissipation terms. For completeness, we also include a small ablation study in Section 5.7 that probes a state-dependent conservative coupling within the same incidence wiring.

As a consequence, we may decompose the vector field as $F(z) = J e + F_{\text{diss}}(e)$ with $e = \nabla H(z)$, where the dissipative part is monotone in energy coordinates in the sense that $\frac{\partial F_{\text{diss}}}{\partial e}(e) = -R(z) \preceq 0$. If dissipation is absent, set $F_{\text{diss}} \equiv 0$ (equivalently $R \equiv 0$). Then $\dot{z} = J e$ and $\dot{H} = e^\top F = e^\top J e = 0$, so the flow conserves energy. In the general case, $\dot{H} = e^\top F = e^\top F_{\text{diss}} = -e^\top R(z) e \leq 0$, so the energy is analytically guaranteed to be nonincreasing along trajectories whenever $R(z) \succeq 0$. External source terms can also be added in the standard port–Hamiltonian manner without altering the incidence wiring.

## 4. MeshFT-Net: Neural Realization of MeshFT

We now instantiate the reduction as an architecture. We can consider two instantiations consistent with the differential form above: **General.** Parameterize a strictly convex storage $H_\theta$ (and a convex dissipation potential $\Psi_\theta$). Here $\Psi_\theta$ is a convex function whose gradient yields the dissi-

---

**Algorithm 1** MeshFT-Net: One-Layer Update

**Inputs:** time step $\Delta t$. fixed $J = \begin{pmatrix} 0 & -D_k^\top \\ D_k & 0 \end{pmatrix}$. learnable SPD $G_\theta$ and PSD $R_\theta$. $z^n = (z_k^n, z_{k+1}^n)$
**Outputs:** $z^{n+1} = (z_k^{n+1}, z_{k+1}^{n+1})$
**(A) Half-damp (in)**
$z_k^{n,-} \leftarrow \exp\left(-\frac{\Delta t}{2} R_{k,\theta}(\{z_k^n, z_{k+1}^n\}) G_k\right) z_k^n$
$z_{k+1}^{n,-} \leftarrow \exp\left(-\frac{\Delta t}{2} R_{k+1,\theta}(\{z_k^n, z_{k+1}^n\}) G_{k+1}\right) z_{k+1}^n$
**(B) Conservative pass (KDK under $J$)**
$z_k^{\text{half}} \leftarrow z_k^{n,-} - \frac{\Delta t}{2} D_k^\top(G_{k+1} z_{k+1}^{n,-})$
$z_{k+1}^{\text{pre}} \leftarrow z_{k+1}^{n,-} + \Delta t \, D_k(G_k z_k^{\text{half}})$
$z_k^{\text{pre}} \leftarrow z_k^{\text{half}} - \frac{\Delta t}{2} D_k^\top(G_{k+1} z_{k+1}^{\text{pre}})$
$z^{\text{pre}} \leftarrow \{z_k^{\text{pre}}, z_{k+1}^{\text{pre}}\}$ (CFLGUARD$(\Delta t)$)
**(C) Half-damp (out)**
$z_k^{n+1} \leftarrow \exp\left(-\frac{\Delta t}{2} R_{k,\theta}(z^{\text{pre}}) G_k\right) z_k^{\text{pre}}$
$z_{k+1}^{n+1} \leftarrow \exp\left(-\frac{\Delta t}{2} R_{k+1,\theta}(z^{\text{pre}}) G_{k+1}\right) z_{k+1}^{\text{pre}}$

---

pative force in the energy variables. With the co-energy $e = \nabla H_\theta(z)$, define the dynamics in energy coordinates by $\dot{z} = J e - \nabla_e \Psi_\theta(e)$, so that $\partial F/\partial e = J - \nabla_e^2 \Psi_\theta(e)$ and $G(z) = \nabla^2 H_\theta(z)$. This covers nonlinear constitutive laws. **Quadratic first-order model.** For efficiency, we use a quadratic storage and a first-order linearization around the current state. With $H_\theta(z) = \frac{1}{2} z^\top G_\theta z$ (degreewise $G_\theta \succ 0$, state-independent), we have $e = G_\theta z$ and $\dot{z} \approx (J - R_\theta(z)) e = (J - R_\theta(z)) G_\theta z$. In experiments in this paper, we adopt this quadratic, state-independent $G_\theta$.

**Fixed vs. Learned.** Under Corollary 3.4, we take $J = \begin{pmatrix} 0 & -D_k^\top \\ D_k & 0 \end{pmatrix}$ and thus *do not train* $J$[1]. The *learnable* components are metrix $G_\theta$ (symmetric positive-definite, SPD) and dissipation $R_\theta$ (positive-semidefinite, PSD).

**Time Stepping (one-layer update).** The integrator is one design choice not a unique consequence of the reduction, other numerically consistent variants can be also used. Here, we advance the state with a Strang splitting (Strang, 1968). One concrete realization is given in Algorithm 1. CFL-GUARD$(\Delta t)$ in Algorithm 1 scales the step (or selects substeps) to satisfy a target CFL condition (Courant et al., 1967; LeVeque & Leveque, 1992). Also, all heavy operations here are sparse matrix–vector products, yielding $O(N)$ time and memory, where N is the total number of degrees of freedom.

**Parameterization.** $G_\theta$ is degreewise SPD, implemented as diagonals (softplus) or small Cholesky blocks, optionally conditioned by permutation-equivariant, orientation-even local MLPs using geometry/material features. $R_\theta(z)$ is PSD (e.g., Rayleigh-type $z \mapsto \gamma(\cdot) G_\theta^{-1} z$). State dependence enters only through $R_\theta(z)$.

**Training.** Given $z^n$, compute the layer output $\hat{z}^{n+1} =$

---

[1] A state-dependent extension keeps the same signed-incidence blocks and learns only local gains $C_k(z) \succ 0$ that modulate them.

MeshFT-Net$_{\Delta t}\big(z^n;\, J,\, G_\theta,\, R_\theta\big)$. Training uses a supervised one–step loss, e.g. $\sum_{k\in\mathcal{I}}\mathrm{Loss}\big(\hat{z}_k^{\,n+1},\, z_k^{\,n+1}\big)$, where $\mathcal{I}$ may be all components or a chosen subset. Optionally, this time-stepping can also be *stacked*, with supervision applied only to the final output composed of sub-step evolution. No PDE–residual terms are used. The inductive bias comes from the fixed interconnection $J$ and the SPD/PSD structure of $G_\theta$ and $R_\theta$.

## 5. Experiments

We evaluate the practical implication of Theorem 3.3 and its corollaries by adopting the induced parameterization: we fix the incidence-wired conservative coupling and learn only the metric operators. We hypothesize that this structural prior preserves long-horizon stability, increases physical fidelity, and improves data efficiency. For comparisons with HNN, we work in canonical variables $(x_k, p_k)$, where $p_k$ is the momentum conjugate to $x^{(k)}$, rather than using flux variables $x_{k+1}$; see Appendix D for details.

**Baselines.** Consider four graph-based simulators, ranging from unconstrained to structure-preserving. All models are trained on the same data under identical training protocols and share same symmetric second-order integrator. To isolate architectural differences, we deliberately chose pedagogically clear, structurally comparable baselines.

**MeshFT-Net.** A structure-preserving model based on Theorem 3.3: the interconnection is fixed by signed incidences, while the metric $G_\theta$ and optional dissipation $R_\theta$ are learned. **MGN.** A structure-agnostic message-passing network that predicts $\dot{z} = v_\phi(z)$ from node/edge features without enforcing physical structure. **MGN-HP (MGN with Hamiltonian Penalty).** An MGN augmented with a learned scalar energy $H_\theta(z)$ and a penalty that aligns $v_\phi(z)$ with the Hamiltonian vector field $X_{H_\theta}(z)$ (in canonical coordinates $X_{H_\theta}(q,p) = (\partial_p H_\theta, -\partial_q H_\theta)$). This preserves MGN's flexibility while nudging it toward conservative dynamics. **HNN.** A Hamiltonian model that learns $H_\theta(z)$ and defines the field by $X_{H_\theta}$. Training minimizes derivative mismatch $\|\dot{z} - X_{H_\theta}(z)\|^2$. We instantiate the common separable form $H(q,p) = U(q) + T(p)$, though nonseparable $H_\theta$ is also compatible.

### 5.1. Analytic Plane-Wave Benchmark

We evaluate each model with periodic 2D plane-wave on regular grids and Delaunay triangulations. Metrics are (i) one-step mean squared error (MSE), (ii) time-series mean squared error (TSMSE) that is the average MSE over the full rollout horizon, and (iii) normalized energy drift over open-loop rollouts. We also vary the training data size. On regular grids (Table 2), MeshFT-Net attains the lowest error and drift, while MGN, MGN-HP, and HNN show orders-

*Table 2.* One-step MSE, normalized energy drift ($\Delta E/E_0$), and TSMSE over the rollout horizon on regular grids. Lower is better and drift closer to 0 is better. All numbers are computed on held-out validation data. Mean over 5 seeds are reported.

| Model | One-step MSE | TSMSE | Energy drift |
|---|---|---|---|
| MeshFT-Net | $\mathbf{1.3 \times 10^{-9}}$ | $\mathbf{9.6 \times 10^{-5}}$ | $\mathbf{1.3 \times 10^{-4}}$ |
| MGN | $1.6 \times 10^{-7}$ | $1.3 \times 10^{-1}$ | $25.9$ |
| MGN-HP | $5.7 \times 10^{-4}$ | $6.1 \times 10^{-1}$ | $16.0$ |
| HNN | $3.5 \times 10^{-8}$ | $3.0 \times 10^{-3}$ | $1.0 \times 10^{-2}$ |

*Table 3.* Dissipative benchmark. One-step MSE is the per-step prediction error. TSMSE is the time-series mean squared error over the rollout. NEE is the normalized energy error relative to the theoretical energy. Lower is better. All numbers are computed on held-out validation data. Means over 3 seeds are reported.

| Model | One-step MSE | TSMSE | NEE |
|---|---|---|---|
| MeshFT-Net | $1.2 \times 10^{-7}$ | $\mathbf{2.1 \times 10^{-2}}$ | $\mathbf{2.1 \times 10^{-2}}$ |
| MGN | $\mathbf{5.2 \times 10^{-8}}$ | $1.7 \times 10^{-1}$ | $2.2$ |
| MGN-HP | $4.5 \times 10^{-4}$ | $1.9$ | $5.0$ |
| HNN | $2.5 \times 10^{-7}$ | $2.2 \times 10^{-1}$ | $4.9 \times 10^{-1}$ |

of-magnitude larger drift. The same ordering holds on Delaunay meshes. Across data sizes (Figs. 2, 5), MeshFT-Net maintains near-zero drift and achieves higher data efficiency relative to baselines. HNN enforces Hamiltonian structure with soft penalties. It improves stability over vanilla MGN, MeshFT-Net's topological interconnection with a symplectic step yields orders of magnitude greater robustness. Implementation details and additional results appear in Appendix D.1.

We also test a Rayleigh-damped setting (amplitude $\propto e^{-\gamma t}$); details appear in Appendix D.2. For HNN, we also learn an explicit Rayleigh damping term same as MeshFT-Net. We report one-step MSE, the time-series mean squared error (TSMSE) over the full rollout horizon, and the *normalized energy error* (NEE) relative to the theoretical energy. As shown in Table 3, MGN attains the lowest one-step error, while MeshFT-Net achieves the best energy fidelity and the lowest TSMSE. These results indicate that fixing the incidence-based interconnection and learning only metric/dissipation captures dissipation most faithfully.

### 5.2. Physics-Consistency Benchmark

Beyond accuracy and energy drift, we ask whether models respect the *physics* on 2D plane waves (periodic grids). We use model-agnostic diagnostics for all baselines: (i) wave speed error, (ii) canonical consistency, (iii) PDE residual, (iv) kinetic–potential equipartition, and (v) momentum conservation. We also run lightweight learning-validity checks: vector-field alignment and amplitude/phase fit. Full definitions appear in Appendix D.3. To broaden coverage of

*Table 4.* Physical-consistency and learning-adequacy diagnostics. Physical-consistency uses $T$=200, $\Delta t$=0.002 (lower is better). PDE residuals are reported as short/long. The short residual is computed over $T$=5 and the long residual over $T$=200. Learning-adequacy uses $T_{\text{short}}$=16 (higher is better for cosine, lower otherwise). All numbers are computed on held-out validation data. Mean over 5 seeds is reported. Best values are shown in bold, and second-best values are underlined.

**(a) Physical-consistency**

| Model | Wave-speed | Canonical | PDE resid. short/long | Equipartition | Momentum |
|---|---|---|---|---|---|
| MeshFT-Net | $\mathbf{2.03 \times 10^{-2}}$ | $\mathbf{9.63 \times 10^{-6}}$ | $\mathbf{3.61 \times 10^{-3}/3.32 \times 10^{-3}}$ | $\mathbf{4.11 \times 10^{-2}}$ | $\mathbf{4.89 \times 10^{-8}}$ |
| MGN | $2.00 \times 10^{-1}$ | $1.02 \times 10^{-3}$ | $3.56 \times 10^{-2}/2.68 \times 10^{-1}$ | $\underline{1.68 \times 10^{-1}}$ | $3.90 \times 10^{-1}$ |
| MGN-HP | $3.91 \times 10^{-1}$ | $1.65 \times 10^{-3}$ | $1.11 \times 10^{-1}/2.21 \times 10^{-1}$ | $2.60 \times 10^{-1}$ | $9.01 \times 10^{-2}$ |
| PI-MGN | $4.84 \times 10^{-1}$ | $15.9$ | $2.48 \times 10^{-2}/6.68 \times 10^{-1}$ | $4.39 \times 10^{-1}$ | $4.99$ |
| HNN | $\underline{8.88 \times 10^{-2}}$ | $\underline{6.49 \times 10^{-5}}$ | $\underline{1.88 \times 10^{-2}}/2.01 \times 10^{-1}$ | $2.01 \times 10^{-1}$ | $1.07$ |
| FNO | $3.08 \times 10^{-1}$ | $6.95 \times 10^{-2}$ | $5.09 \times 10^{-2}/5.81 \times 10^{-1}$ | $2.60 \times 10^{-1}$ | $1.65$ |
| GraphCON | $2.11 \times 10^{-1}$ | $2.42 \times 10^{-2}$ | $1.57 \times 10^{-1}/2.80 \times 10^{-1}$ | $1.79 \times 10^{-1}$ | $3.00 \times 10^{-1}$ |

**(b) Learning-adequacy**

| Model | VF cosine ($\uparrow$) | VF $L_2$ | Short roll MSE | Amp err | Phase err (deg) |
|---|---|---|---|---|---|
| MeshFT-Net | $\mathbf{9.999\,96 \times 10^{-1}}$ | $\mathbf{3.03 \times 10^{-3}}$ | $\underline{9.36 \times 10^{-2}}$ | $\mathbf{1.25 \times 10^{-2}}$ | $\mathbf{8.06 \times 10^{-1}}$ |
| MGN | $9.998\,82 \times 10^{-1}$ | $2.85 \times 10^{-2}$ | $1.06 \times 10^{-1}$ | $5.05 \times 10^{-2}$ | $2.47$ |
| MGN-HP | $9.993\,53 \times 10^{-1}$ | $2.80 \times 10^{-2}$ | $2.11 \times 10^{-1}$ | $1.66 \times 10^{-1}$ | $5.99$ |
| PI-MGN | $9.355\,56 \times 10^{-1}$ | $2.96 \times 10^{-1}$ | $2.98 \times 10^{-1}$ | $6.93 \times 10^{-2}$ | $13.9$ |
| HNN | $\underline{9.999\,22 \times 10^{-1}}$ | $\underline{1.14 \times 10^{-2}}$ | $\mathbf{6.34 \times 10^{-2}}$ | $\underline{2.14 \times 10^{-2}}$ | $\underline{1.05}$ |
| FNO | $9.998\,21 \times 10^{-1}$ | $2.32 \times 10^{-2}$ | $1.26 \times 10^{-1}$ | $8.29 \times 10^{-2}$ | $3.98$ |
| GraphCON | $9.987\,79 \times 10^{-1}$ | $4.62 \times 10^{-2}$ | $1.96 \times 10^{-1}$ | $1.26 \times 10^{-1}$ | $4.60$ |

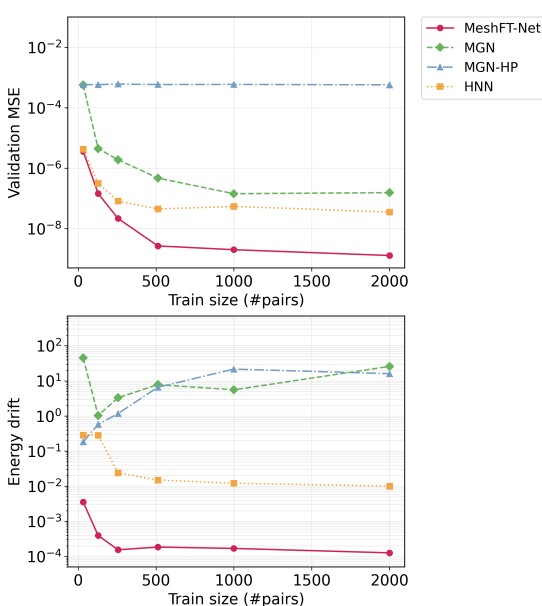

*Figure 2.* Relationship between the size of training dataset and one-step MSE (top) and rollout energy drift (bottom) for grid mesh.

baselines, we also add a neural operator (FNO) (Li et al., 2021), a long-range graph simulator (GraphCON) (Rusch et al., 2022), and a physics-informed MeshGraphNet (PI-MGN) (Würth et al., 2024) with MGN, MGN-HP, and HNN.

Table 4 (a) shows that MeshFT-Net is the most physically faithful across the diagnostics. It attains the smallest wave-speed error, an essentially exact canonical relation, the low-

est PDE residuals, the closest equipartition, and near-zero momentum change, outperforming all baselines, often by orders of magnitude. Momentum is preserved without an explicit constraint because MeshFT-Net enforces the interface action–reaction structure implied by (O), whereas unconstrained baselines need not. The additional baselines follow the same pattern. FNO and GraphCON can keep short-horizon errors reasonable but exhibit larger long-horizon residuals and momentum drift, while PI-MGN improves some accuracy terms but degrades canonical consistency and long-horizon residuals. The learning-adequacy checks in Table 4 (b) are consistent with this picture. MeshFT-Net achieves the best vector-field alignment, the second-smallest short-horizon rollout error, and the most accurate amplitude and phase recovery. A soft Hamiltonian penalty, as in HNN, can improve some diagnostics, but MeshFT-Net remains more robust overall. These results support the effectiveness for yielding physically consistent predictions of a principle-driven approach in which through the reduction theorem the topology-wired interconnection is fixed and only the metric factors are learned.

### 5.3. OOD Generalization

We also evaluate OOD generalization on periodic 2D waves under three shifts such as (i) frequency, (ii) resolution (coarse to fine), and (iii) parameter (wave speed). Metrics are TSMSE and normalized energy drift after a fixed-horizon rollout; details and full results in Table 11 of Ap-

*Table 5.* OOD generalization under Frequency, Wave speed, and Resolution shifts (lower is better). Columns report TSMSE and normalized energy drift, computed on held-out validation sets. Values are means over 3 seeds. Bold indicates the best and underline the second best in each column. Entries shown as $> 100$ exceeded the evaluation cap (diverged). Scientific notation in this table uses e-format for compactness. Full results are in Appendix D.4.

| Model | Frequency | | Wave Speed | | Resolution | |
|---|---|---|---|---|---|---|
| | TSMSE | Drift | TSMSE | Drift | TSMSE | Drift |
| MeshFT-Net | **1.8e-1** | **5.1e-3** | 7.1e-1 | **1.7e-1** | **5.1e-2** | **2.9e-3** |
| MGN | 9.0 | $> 100$ | 1.7 | 25.5 | 5.3e-1 | 3.4 |
| MGN-HP | 6.2e-1 | 1.6e-1 | **5.9e-1** | 2.5e-1 | 5.8e-1 | 5.6 |
| PI-MGN | $> 100$ | $> 100$ | $> 100$ | $> 100$ | $> 100$ | $> 100$ |
| HNN | 1.54 | 4.5e-1 | 8.0e-1 | 2.4e-1 | 3.6e-1 | 2.0 |
| FNO | 1.94 | 2.7 | 9.6e-1 | 1.6 | $> 100$ | $> 100$ |
| GraphCON | 69.0 | 96.4 | $> 100$ | $> 100$ | 3.2e-1 | 7.1e-1 |

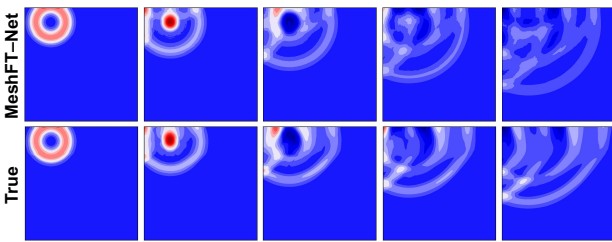

*Figure 3.* Pressure snapshots at equal time steps ordered right to left. The rightmost frame is the initial state (shared colormap).

pendix D.4. This tests whether inductive bias supports accurate, physically consistent rollouts under resolution and physical parameters shifts. We include baselines MGN, MGN-HP, PI-MGN, HNN, FNO, and GraphCON. Across all OOD shifts in Table 5, MeshFT-Net delivers the lowest energy drift overall and the best TSMSE on *Frequency* and *Resolution*. Several baselines diverge under certain shifts ($> 100$), whereas MeshFT-Net remains stable and accurate across discretization, and parameter changes. Importantly, together with the one-step MSE shown in Table 11, these results show that strong local generalization does not necessarily translate into faithful long-horizon dynamics or energy stability. Models with competitive one-step MSE can still exhibit larger TSMSE and/or drift over rollouts. Overall, MeshFT-Net fixes the incidence-based interconnection and learns only metric/dissipation factors, yielding robust extrapolation across resolution and physical parameters shifts.

### 5.4. Acoustic Scattering Benchmark from *The Well*

To assess transfer beyond synthetic data, we evaluate MeshFT-Net on a subset of *The Well*—Acoustic Scattering (Ohana et al., 2024; Mandli et al., 2016) which is near-Hamiltonian but includes discontinuous media and open/reflective boundaries. We train with one-step teacher forcing for the pressure field. This probes whether a

*Table 6.* Nonlinear, dissipative, and heterogeneous benchmarks. TSMSE is reported on held-out OOD rollouts. All values are means over 3 seeds. Lower is better. Best values are shown in bold, and second-best values are underlined. Entries shown as $> 100$ exceeded the display cap or yielded non-finite rollout statistics. For (a), FNO is omitted because it is grid-based. Full experimental details, scalability results, and qualitative rollouts are provided in Appendix D.6.

**(a) Irregular heterogeneous elasticity**

| Model | TSMSE | Momentum variation |
|---|---|---|
| MeshFT-Net | $4.91 \times 10^{-3}$ | $3.09 \times 10^{-7}$ |
| MGN | $2.01 \times 10^{-1}$ | 1.51 |
| MGN-HP | $6.88 \times 10^{-2}$ | 1.11 |
| HNN | $2.95 \times 10^{-2}$ | $1.85 \times 10^{-1}$ |
| GraphCON | $> 100$ | $> 100$ |

**(b) Damped nonlinear lattice oscillator**

| Model | TSMSE | Energy rel. MAE |
|---|---|---|
| MeshFT-Net | $2.54 \times 10^{-4}$ | $8.70 \times 10^{-3}$ |
| MGN | $> 100$ | $> 100$ |
| MGN-HP | $> 100$ | $> 100$ |
| HNN | $1.15 \times 10^{-2}$ | $3.01 \times 10^{-2}$ |
| FNO | $1.70 \times 10^{-1}$ | $4.32 \times 10^{-1}$ |
| GraphCON | $9.19 \times 10^{-4}$ | $1.98 \times 10^{-2}$ |

topology-fixed interconnection that preserves structure also maintains fidelity for more realistic data. The details of this experiment appear in Appendix D.5. Fig. 3 shows the snapshots of predicted and ground-truth pressure fields on the validation set, using a common colormap. MeshFT-Net closely matches wavefront position and curvature. Differences are limited to slight smoothing of high-frequency details and reduced peak contrast, while boundary reflections remain consistent. These visuals agree with the quantitative findings of low drift and correct dispersion.

### 5.5. Nonlinear and Heterogeneous Benchmarks

We next evaluate the MeshFT-Net parameterization in more general cases involving irregular geometry, heterogeneous constitutive response, nonlinear dynamics, and dissipation. The conservative interconnection remains fixed by signed incidences, while geometry- and material-dependent effects are represented through learned metric-dependent factors. Thus, MeshFT-Net keeps the signed-incidence sparsity and orientation structure of $J$ fixed throughout these evaluations. Table 6 reports representative results. Full results, implementation details, scalability measurements, and qualitative visualizations are provided in Appendix D.6.

The irregular heterogeneous-elasticity benchmark uses a bounded deformable body with a hole, an irregular mesh, and geometry-dependent constitutive response. As shown in Table 6 (a), MeshFT-Net attains the lowest rollout error among the evaluated baselines and keeps total momentum variation at the $10^{-7}$ scale. This supports that the signed-

*Table 7.* Topology–metric separation ablation on the heterogeneous-elasticity benchmark. Metrics are means over 3 seeds. Lower is better. Best values are shown in bold, and second-best values are underlined.

| Variant | TSMSE | Energy rel. MAE | Mom. var. |
|---|---|---|---|
| MeshFT-Net | $\mathbf{4.83 \times 10^{-3}}$ | $\mathbf{5.32 \times 10^{-2}}$ | $\mathbf{2.67 \times 10^{-7}}$ |
| w/o metric | $1.50 \times 10^{-2}$ | $9.99 \times 10^{-2}$ | $\underline{2.78 \times 10^{-7}}$ |
| w/o topology-fixed | $\underline{1.41 \times 10^{-2}}$ | $\underline{6.78 \times 10^{-2}}$ | $1.26 \times 10^{-2}$ |

incidence conservative wiring remains effective beyond periodic wave benchmarks, while the learned metric-dependent factors represent spatial heterogeneity. In addition, we assess scalability and qualitative behavior on the same heterogeneous deformable-body setting. Full results are reported in Appendix D.6. Also, the damped nonlinear lattice benchmark shown in Table 6 (b) clarifies the dissipative side of the reduction. MeshFT-Net tracks the dissipative energy decay more accurately than the evaluated baselines even when the system is nonlinear, with lower rollout error and lower energy relative mean absolute error (Energy rel. MAE). This shows that the same topology-fixed interconnection with learned metric-dependent dissipative factors also captures well nonlinear dissipative dynamics.

### 5.6. Topology–Metric Separation Ablation

We isolate the practical roles of topological and metric factors of MeshFT-Net using the heterogeneous-elasticity benchmark. The full model uses topology-fixed signed-incidence conservative wiring together with a learned metric. We compare it with two ablations: one that restricts the metric to a spatially uniform metric while keeping the conservative wiring fixed, and one that breaks the topology-fixed conservative wiring while retaining local metric learning.

Table 7 shows a clear separation of roles. Restricting the heterogeneous metric degrades rollout accuracy and energy fidelity, while momentum variation (Mom. var.) remains at the $10^{-7}$ scale because the signed-incidence wiring is unchanged. In contrast, breaking the topology-fixed conservative wiring leads to a much larger momentum variation. This behavior is consistent with the reduction theorem: metric-dependent factors control constitutive scales and energy fidelity, whereas signed-incidence conservative wiring guarantee the symmetric structure underlying momentum preservation. Implementation details and additional results are provided in Appendix D.8 and complementary principle-level ablations are provided in Appendix D.9.

### 5.7. State-Dependent Conservative-Coupling Ablation

To isolate the role of state-dependent conservative coupling inside MeshFT-Net, we run a controlled nonlinear experiment in which conservative coupling strengths vary with the current state while the signed-incidence wiring is kept

*Table 8.* Controlled state-dependent conservative-coupling ablation. We report TSMSE over a long rollout and relative energy drift under the reference energy induced by $G_{\text{true}}$. Metrics are means over 3 seeds. Lower is better. Best values are shown in bold, and second-best values are underlined.

| Variant | TSMSE | Energy drift |
|---|---|---|
| Fixed-$J$ + Diag-$G$ | $4.52 \times 10^{-5}$ | $1.15 \times 10^{-1}$ |
| Fixed-$J$ + Full-$G$ | $3.28 \times 10^{-5}$ | $\underline{2.79 \times 10^{-2}}$ |
| $z$-dependent-$J$ + Diag-$G$ | $\underline{6.77 \times 10^{-6}}$ | $\mathbf{2.49 \times 10^{-2}}$ |
| $z$-dependent-$J$ + Full-$G$ | $\mathbf{6.17 \times 10^{-6}}$ | $3.01 \times 10^{-2}$ |

fixed. The purpose is to experimentally clarify how nonlinear conservative variation is shared between the interconnection and metric sides. We generate trajectories from a structure-reduced conservative system $\dot{z} = J_{\text{true}}(z)G_{\text{true}}z$ with a fixed diagonal energy metric $G_{\text{true}}$ (details in Appendix D.7). We then vary two modeling choices: state-dependent versus state-independent $J$, and channel-wise diagonal versus full cross-channel $3 \times 3$ metric $G$. Table 8 shows that allowing $z$-dependence in $J$ reduces TSMSE and energy drift, consistent with capturing state-dependent coupling variation through incidence-wired gains. A full cross-channel $G$ can *partly* mitigate the mismatch when $J$ is state-independent, but it does not match the benefit of modeling the coupling variation directly.

## 6. Conclusion

We established *Mesh Field Theory (MeshFT)* by proving a principle-driven *local* Jacobian reduction for mesh dynamics. Under locality, permutation equivariance, orientation consistency, and an energy balance or dissipation inequality, admissible dynamics admit a pointwise port–Hamiltonian factorization in which the conservative coupling has topology-fixed wiring, while geometry, material response, and dissipation enter only through local metric-dependent operators. The novelty is not to posit a port–Hamiltonian template, but to *deduce* what is structurally fixed versus learnable from the physical principles on meshes, thereby making the topology–metric split an identified consequence rather than an assumption.

Guided by this reduction, we designed MeshFT-Net to hardwire the conservative wiring and learn only local metric and dissipation operators. Across analytic datasets and acoustic data from *The Well*, physics-consistency tests, and OOD validation, MeshFT-Net produces accurate long-horizon rollouts with strong physical fidelity, improved data efficiency, and robust extrapolation across resolution and parameter shifts. Consequently, MeshFT takes a first step toward a principle-driven methodology for constructing learning-based simulators on meshes by relying the reduction theorem to deduce the topology-wired conservative structure and confine learning to metric and dissipation.

## Acknowledgements

This work was supported by JSPS KAKENHI Grant Numbers JP23K19968, JP26K21327, JP22H05106 and JP26H02498.

## Impact Statement

This paper presents work whose goal is to advance the field of Machine Learning. There are many potential societal consequences of our work, none which we feel must be specifically highlighted here.

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

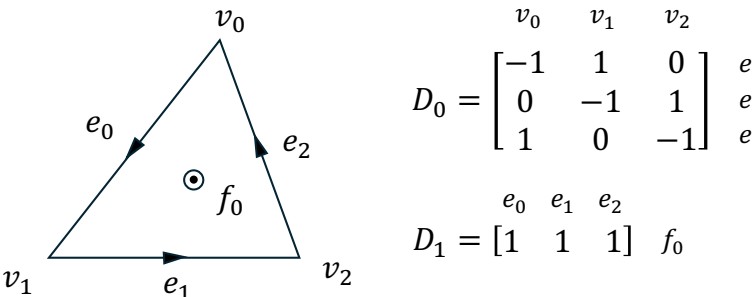

*Figure 4.* A single oriented triangle and its coboundary matrices. Here $N_i$ are vertex labels, $E_i$ are oriented edge labels, and $F_0$ is the oriented face; these labels should not be confused with $n_k$, the number of $k$-cells. Rows of $D_0$ correspond to oriented edges and columns correspond to vertices. Each row has $-1$ at the tail and $+1$ at the head of the corresponding oriented edge. The row of $D_1$ corresponds to the oriented face and columns correspond to oriented edges. In this orientation, all edge orientations agree with the boundary orientation of the face, so $D_1 = [1\ 1\ 1]$. This toy example has $D_0 \in \mathbb{R}^{3\times 3}$ and $D_1 \in \mathbb{R}^{1\times 3}$, but in general $D_k$ is rectangular. The notation $D_k^\top$ always denotes the matrix transpose, not an inverse.

## A. Concrete Examples Coboundary Operators

We give a small example to make the notation used in the main text explicit. Let $n_k$ denote the number of oriented $k$-cells and let $C^k \simeq \mathbb{R}^{n_k}$ be the space of $k$-cochains. A 0-cochain assigns one scalar to each node, a 1-cochain assigns one scalar to each oriented edge, and a 2-cochain assigns one scalar to each oriented face. After fixing an orientation and an ordering of the cells, the coboundary $D_k : C^k \to C^{k+1}$ is represented by a sparse signed-incidence matrix $D_k \in \mathbb{R}^{n_{k+1}\times n_k}$.

For the oriented triangle shown in Fig. 4, we have three vertices, three oriented edges, and one oriented face, hence $n_0 = 3$, $n_1 = 3$, and $n_2 = 1$. With vertex values $q = (q_0, q_1, q_2)^\top \in C^0$ and edge values $u = (u_0, u_1, u_2)^\top \in C^1$, the coboundary matrices can be written as

$$D_0 = \begin{bmatrix} -1 & 1 & 0 \\ 0 & -1 & 1 \\ 1 & 0 & -1 \end{bmatrix}, \qquad D_1 = \begin{bmatrix} 1 & 1 & 1 \end{bmatrix}.$$

Therefore

$$D_0 q = \begin{bmatrix} q_1 - q_0 \\ q_2 - q_1 \\ q_0 - q_2 \end{bmatrix}, \qquad D_1 u = u_0 + u_1 + u_2.$$

Thus $D_0$ sends node values to oriented edge differences, while $D_1$ sums oriented edge quantities around the face. In particular, $D_1 D_0 = 0$, which means that the oriented sum of a discrete gradient around a closed face is zero.

The transpose acts in the reverse direction. For example,

$$D_0^\top u = \begin{bmatrix} -u_0 + u_2 \\ u_0 - u_1 \\ u_1 - u_2 \end{bmatrix},$$

which accumulates signed edge quantities back to vertices. This is why $D_k^\top$ appears in the conservative interconnection.

For a node–edge state $z = (z_0, z_1) \in C^0 \oplus C^1$ with co-energy $e = (e_0, e_1)$, the incidence-wired conservative part used in MeshFT is

$$\begin{bmatrix} \dot{z}_0 \\ \dot{z}_1 \end{bmatrix} = \underbrace{\begin{bmatrix} 0 & -D_0^\top \\ D_0 & 0 \end{bmatrix}}_{J} \begin{bmatrix} e_0 \\ e_1 \end{bmatrix}.$$

The equation $\dot{z}_1 = D_0 e_0$ sends node co-energy differences to oriented edges, while $\dot{z}_0 = -D_0^\top e_1$ sends edge co-energies back to the incident vertices with opposite signs at the two endpoints. The block matrix is skew-symmetric, so the conservative part does no net work:

$$e^\top J e = e_0^\top(-D_0^\top e_1) + e_1^\top D_0 e_0 = 0.$$

This is the elementary mechanism behind the energy-preserving topology-wired coupling.

# B. Proof and Technical Details to Theorem 3.3

We adopt the notation of the main text and work in the local energy coordinates used in Theorem 3.3. That is, near any $z$ where the Jacobian exists, we write $e = \nabla H(z)$ with $G(z) := \frac{\partial e}{\partial z}(z) \succ 0$, for a (local) strictly convex energy $H$, and then $G(z) = \nabla^2 H(z)$. The dynamics split as $\dot{z} = F(z) = F_{\text{con}}(z) + F_{\text{diss}}(z)$ with conservative power balance $e^\top F_{\text{con}}(z) = 0$ and passivity $e^\top F_{\text{diss}}(z) \le 0$ (no sources).

**Orientation/Sign Conventions.** Let $\mathcal{K}$ be an oriented cell complex with cochain spaces $\{C^k\}_{k=0}^d$ and signed coboundaries $D_k : C^k \to C^{k+1}$ satisfying $D_{k+1}D_k = 0$. Degree-wise flips are encoded by $\rho = \text{diag}(\rho_0, \dots, \rho_d)$, where each $\rho_k$ is diagonal with $\pm 1$ entries (orientation gauge). Consistent flips leave incidences invariant, $\rho_{k+1}D_k\rho_k = D_k$, whereas single-degree flips change sign: $D_k\rho_k = -D_k$ and $\rho_{k+1}D_k = -D_k$. Orientation covariance (O) is expressed by

$$e(\rho z) = \rho\, e(z), \quad F(\rho z) = \rho\, F(z), \tag{1}$$

so scalar quantities (energy, power) are gauge-invariant while flux-like quantities co-transform with their carriers. These conventions will be used repeatedly in the proofs below. This also implies $H(\rho z) = H(z)$.

## B.1. Local Symmetry Basis and Orientation Covariance

We first formalize the structure forced by (L), (P), and orientation covariance (O).

**Lemma B.1** (Local, permutation-equivariant linear basis)**.** *Let* $T : C^{k-1} \times C^k \times C^{k+1} \to C^k$ *be linear,* interface-local *(depends only on cells incident to the output $k$-cell), and permutation-equivariant (with permutations acting within each degree/type class). Then $T$ decomposes as*

$$T(x_{k-1}, x_k, x_{k+1}) = a\, x_k\ +\ b\, A_k x_k\ +\ \alpha\, D_{k-1} x_{k-1}\ -\ \beta\, D_k^\top x_{k+1}, \tag{2}$$

*where $A_k$ is the* unsigned *adjacency of $k$-cells on $\mathcal{K}$, and coefficients $(a, b, \alpha, \beta)$ are label-independent scalars.*

*Proof.* (L) implies that the output on any $k$-cell can only use: itself, adjacent $k$-cells (sharing an interface), its $(k-1)$ faces, and its $(k+1)$ cofaces. From (P), relabeling cells within a degree/type must only relabel the output, so values within the same relation type (self, $k$-neighbors, faces, cofaces) must share one common weight. With linearity, the only degree-compatible linear maps supported on this incident neighborhood are the identity on $C^k$, the unsigned $k$–$k$ adjacency $A_k : C^k \to C^k$, the boundary $D_{k-1} : C^{k-1} \to C^k$, and the coboundary $-D_k^\top : C^{k+1} \to C^k$.

Hence $T$ is the stated linear combination. Any other term would either use non-incident cells (violating locality) or assign different weights within a relation type (violating permutation equivariance). $\square$

**Lemma B.2** (Orientation covariance rules out same-degree terms)**.** *From* (O)*, in the decomposition of Lemma B.1, we must have $a = b = 0$. Consequently,*

$$T(x_{k-1}, x_k, x_{k+1})\ =\ \alpha\, D_{k-1} x_{k-1}\ -\ \beta\, D_k^\top x_{k+1}. \tag{3}$$

*Proof.* By (O), if we change the sign convention only at degree $k$, then the $k$-oriented output flips sign accordingly:

$$\rho_k^{-1}\, T(x_{k-1}, \rho_k x_k, x_{k+1})\ =\ -T(x_{k-1}, x_k, x_{k+1}) \quad \text{for all inputs.}$$

Evaluate the basis terms from Lemma B.1 under this operation. Since $A_k$ is unsigned and depends only on incidence, it commutes with $\rho_k$. Hence

$$\rho_k^{-1}(a\, \rho_k x_k) = a\, x_k, \qquad \rho_k^{-1}(b\, A_k\, \rho_k x_k) = b\, A_k x_k,$$

while the cross-degree terms change sign on the $k$-side:

$$\rho_k^{-1}(D_{k-1} x_{k-1}) = -\, D_{k-1} x_{k-1}, \qquad \rho_k^{-1}(D_k^\top x_{k+1}) = -\, D_k^\top x_{k+1}.$$

Therefore the left-hand side equals $a\, x_k + b\, A_k x_k - \alpha\, D_{k-1} x_{k-1} + \beta\, D_k^\top x_{k+1}$. Since this must be the negative of $T(x_{k-1}, x_k, x_{k+1})$ for all inputs, the within-degree part must vanish, i.e., $a = b = 0$.

Finally, (O) also requires consistency under simultaneous flips across degrees. Using the standard sign behavior,

$$\rho_k^{-1} D_{k-1} \rho_{k-1} = D_{k-1}, \qquad \rho_k^{-1} D_k^\top \rho_{k+1} = D_k^\top,$$

so the remaining cross-degree terms satisfy (O). This yields the stated form. $\square$

**Note on Same-Degree Adjacency.** If we split $A_k$ into lower/upper $k$–$k$ adjacencies (sharing a $(k{-}1)$- or a $(k{+}1)$-cell) with independent weights, (O) still forces both weights to be zero. Indeed, both adjacency operators commute with the degree-$k$ flip, while (O) requires $T$ to change sign under that flip (after compensating the output sign). Hence Lemma B.2 and its consequences remain unchanged.

## B.2. Skew/Dissipative Split from Energy Balance

The next lemma formalizes, at the differential level, the split implied by the incremental energy assumptions (E).

**Lemma B.3** (Passivity induces a skew/dissipative representation). *Let $e = \nabla H(z)$ be local energy coordinates in a neighborhood of $z$, with Jacobian $G(z) = \frac{\partial e}{\partial z}(z) \succ 0$. Assume $F$ is differentiable at $z$ and that the incremental energy conditions hold in a neighborhood of $z$, i.e., for all $z_1, z_2$ near $z$ with $e_i = \nabla H(z_i)$,*

$$(e_1 - e_2)^\top\big(F_{\mathrm{con}}(z_1) - F_{\mathrm{con}}(z_2)\big) = 0, \qquad (e_1 - e_2)^\top\big(F_{\mathrm{diss}}(z_1) - F_{\mathrm{diss}}(z_2)\big) \le 0. \tag{4}$$

*Then there exist matrices $J(z)$ and $R(z)$ with $J(z)^\top = -J(z)$ and $R(z) \succeq 0$ such that*

$$\frac{\partial F}{\partial z}(z) = \big(J(z) - R(z)\big)G(z). \tag{5}$$

*Proof.* Fix $z$ and a direction $\delta z$. Set $z_1 = z + \varepsilon\,\delta z$, $z_2 = z$ and define

$$\phi_{\mathrm{con}}(\varepsilon) := (e_1 - e_2)^\top\big(F_{\mathrm{con}}(z_1) - F_{\mathrm{con}}(z_2)\big), \quad \phi_{\mathrm{diss}}(\varepsilon) := (e_1 - e_2)^\top\big(F_{\mathrm{diss}}(z_1) - F_{\mathrm{diss}}(z_2)\big). \tag{6}$$

By the incremental energy conditions, $\phi_{\mathrm{con}}(\varepsilon) \equiv 0$ and $\phi_{\mathrm{diss}}(\varepsilon) \le 0$ for small $\varepsilon$. Differentiability of $\nabla H$ gives

$$e_1 - e_2 = \varepsilon\,G(z)\,\delta z + r_e(\varepsilon), \qquad \|r_e(\varepsilon)\| = o(\varepsilon), \tag{7}$$

and differentiability of $F_{\mathrm{con}}$ and $F_{\mathrm{diss}}$ yields

$$F_{\mathrm{con}}(z_1) - F_{\mathrm{con}}(z_2) = \varepsilon\,A_{\mathrm{con}}(z)\,\delta z + r_{\mathrm{con}}(\varepsilon), \quad F_{\mathrm{diss}}(z_1) - F_{\mathrm{diss}}(z_2) = \varepsilon\,A_{\mathrm{diss}}(z)\,\delta z + r_{\mathrm{diss}}(\varepsilon), \tag{8}$$

with $A_{\mathrm{con}}(z) = \frac{\partial F_{\mathrm{con}}}{\partial z}(z)$, $A_{\mathrm{diss}}(z) = \frac{\partial F_{\mathrm{diss}}}{\partial z}(z)$, and $\|r_{\mathrm{con}}(\varepsilon)\|, \|r_{\mathrm{diss}}(\varepsilon)\| = o(\varepsilon)$. Let $\delta e := G(z)\delta z$. Since $G(z)$ is nonsingular, $\delta e$ ranges over all directions.

Substituting into $\phi_{\mathrm{con}}(\varepsilon)$ and $\phi_{\mathrm{diss}}(\varepsilon)$ gives

$$\phi_{\mathrm{con}}(\varepsilon) = \varepsilon^2\,\delta e^\top\Big(A_{\mathrm{con}}(z)\,G(z)^{-1}\Big)\delta e + o(\varepsilon^2), \tag{9}$$

$$\phi_{\mathrm{diss}}(\varepsilon) = \varepsilon^2\,\delta e^\top\Big(A_{\mathrm{diss}}(z)\,G(z)^{-1}\Big)\delta e + o(\varepsilon^2). \tag{10}$$

Dividing by $\varepsilon^2$ and letting $\varepsilon \to 0$ yields, for all $\delta e$,

$$\delta e^\top K_{\mathrm{con}}(z)\,\delta e = 0, \qquad \delta e^\top K_{\mathrm{diss}}(z)\,\delta e \le 0, \tag{11}$$

where we define $K_{\mathrm{con}}(z) := A_{\mathrm{con}}(z)G(z)^{-1}$ and $K_{\mathrm{diss}}(z) := A_{\mathrm{diss}}(z)G(z)^{-1}$. Therefore $\mathrm{Sym}(K_{\mathrm{con}}(z)) = 0$ and $\mathrm{Sym}(K_{\mathrm{diss}}(z)) \preceq 0$. Set

$$R(z) := -\,\mathrm{Sym}\big(K_{\mathrm{diss}}(z)\big) \succeq 0, \qquad J(z) := K_{\mathrm{con}}(z) + \mathrm{Skew}\big(K_{\mathrm{diss}}(z)\big), \tag{12}$$

so that $J(z)^\top = -J(z)$ and $K_{\mathrm{con}}(z) + K_{\mathrm{diss}}(z) = J(z) - R(z)$. Finally,

$$\frac{\partial F}{\partial z}(z) = A_{\mathrm{con}}(z) + A_{\mathrm{diss}}(z) = \big(K_{\mathrm{con}}(z) + K_{\mathrm{diss}}(z)\big)G(z) = \big(J(z) - R(z)\big)G(z),$$

as claimed. $\qquad\square$

### B.3. Incidence wiring of the conservative interconnection

**Lemma B.4** (Incidence wiring of the conservative interconnection). *Assume* (L)*,* (P)*, and* (O)*. Let* $K : C^k \oplus C^{k+1} \to C^k \oplus C^{k+1}$ *be linear, interface-local, and permutation-equivariant within each degree/type class, and assume* $K^\top = -K$ *and orientation covariance. Then, up to cochain ordering and an orientation gauge, for each adjacent pair* $(k, k{+}1)$ *the off-diagonal blocks have signed-incidence wiring and can be written as*

$$K_{k,k+1} = -D_k^\top C_k, \quad K_{k+1,k} = C_k D_k,$$

*for some diagonal operator* $C_k$ *with positive diagonal entries.*

*Proof.* By Lemma B.1, any degree-$k$ output can involve only $\{\mathrm{Id}, A_k, D_{k-1}, -D_k^\top\}$. By Lemma B.2, orientation covariance rules out the same-degree terms Id and $A_k$, so only cross-degree incidence terms remain. Skew-symmetry forces the $(k, k{+}1)$ and $(k{+}1, k)$ blocks to be negatives of each other, hence they share the same per-incidence weights. By locality and permutation equivariance, these weights can depend only on the local incidence type and must be shared across relabelings within the degree/type class. Finally, by an orientation gauge choice one may take the diagonal entries to be positive. $\qquad\square$

### B.4. Proof of Theorem 3.3

Fix a point $z$ where the Jacobian exists and work in the local energy coordinates $e = \nabla H(z)$ with $G(z) = \frac{\partial e}{\partial z}(z) \succ 0$. By Lemma B.3,

$$\frac{\partial F}{\partial z}(z) = \big(J(z) - R(z)\big)G(z), \qquad J(z)^\top = -J(z), \ \ R(z) \succeq 0.$$

It remains to identify the conservative wiring implied by (L), (P), and (O). Applying Lemma B.4 pointwise to the linear map $K := J(z)$ yields, up to cochain ordering and an orientation gauge, the signed-incidence block form

$$J_{k,k+1}(z) = -D_k^\top C_k(z), \qquad J_{k+1,k}(z) = C_k(z)D_k,$$

where we write $C_k(z)$ to emphasize that the diagonal gain may vary with $z$ through the dependence of $J(z)$ on the state. This is exactly the signed-incidence wiring statement in Theorem 3.3.

**Incidence normalization (proof for Corollary 3.4).** If the incidence gain is state-independent for the pair $(k, k{+}1)$ so that $C_k(z) \equiv C_k$, define the constant rescaling $S = \mathrm{diag}(I, C_k)$ on $C^k \oplus C^{k+1}$ and the rescaled field $\tilde{F}(\tilde{z}) = S^{-1}F(S\tilde{z})$ with $z = S\tilde{z}$. Then the induced interconnection has pure signed-incidence blocks $\tilde{J}_{k,k+1} = -D_k^\top$ and $\tilde{J}_{k+1,k} = D_k$, with the metric-side factors transforming as $\tilde{G}(\tilde{z}) = S^\top G(S\tilde{z})S$ and $\tilde{R}(\tilde{z}) = S^{-1}R(S\tilde{z})S^{-\top}$.

**Remark.** Theorem 3.3 allows state-dependent conservative coupling through the diagonal gains $C_k(z)$ while keeping the signed-incidence wiring fixed. If $C_k(z) \equiv C_k$ is state-independent (e.g., geometry or material only), one can pass to an equivalent incidence-normalized representative by a fixed rescaling as in Corollary 3.4. If $C_k$ depends on $z$, such a fixed rescaling cannot remove the dependence in general, and the interconnection remains state-dependent.

**Generality and Sharpness.** The reduction is architecture-independent and applies to any vector field $F$ satisfying (L), (P), (O), and (E). At any $z$ where the Jacobian exists it yields the pointwise factorization $\partial F/\partial z = (J(z) - R(z))\,G(z)$, where the off-diagonal blocks of $J(z)$ have signed-incidence wiring. Under state-independent incidence gains, this further admits a state-independent interconnection and, after incidence normalization, a constant purely incidence-based wiring.

The hypotheses are also essentially minimal. Dropping (O) allows same-degree, orientation-insensitive terms such as the identity $I$ and unsigned adjacency $A_k$. Dropping (P) permits interface-wise heterogeneity within a degree/type class, breaking equivariance and the clean incidence assembly. Dropping (L) admits nonlocal couplings beyond the incidence neighborhood. Relaxing (E) forfeits the skew–dissipative $(J-R)G$ split. Each assumption therefore excludes a concrete failure mode, and together they separate what is fixed by topology from what remains learnable on the metric side. An empirical ablation study is provided in Appendix D.9.

**Linear-Media Corollary.** If, in addition, $H(z) = \frac{1}{2}z^\top M^{-1}z$ with constant $M \succ 0$ and $F_{\mathrm{diss}}(z) = -R\,e$ with state-independent $R^\top = R \succeq 0$, then $G \equiv M^{-1}$ and the dynamics reduce to

$$\dot{z} = (J - R)\,M^{-1}z, \quad J^\top = -J, \ \ R^\top = R \succeq 0, \tag{13}$$

with $(M, R)$ carrying the metric and dissipation.

## C. Electromagnetism Example

As an intuitive example, we revisit electromagnetism though the lens of Theorem 3.3.

### C.1. Recovering Source-Free Maxwell from Topology-Fixed Incidence Structure

On an oriented mesh with edge–face incidence $D_1$, the constitutive maps are the (degree–wise) SPD Hodge stars

$$H = \star_{\mu^{-1}} B, \qquad E = \star_{\varepsilon^{-1}} D, \tag{14}$$

where $B$ is the magnetic flux density and $H$ the magnetic field intensity. Also, $D$ is the electric flux density and $E$ the electric field. The operator $\star_\kappa$ denotes the (discrete) Hodge star composed with the material tensor $\kappa$ (e.g., $\mu^{-1}$, $\varepsilon^{-1}$).

So, stacking $z = (B, D)$ and $e = (H, E)$, one has

$$e = \underbrace{\begin{pmatrix} \star_{\mu^{-1}} & 0 \\ 0 & \star_{\varepsilon^{-1}} \end{pmatrix}}_{G_\theta^{-1}} z. \tag{15}$$

The conservative Maxwell update (no sources) is

$$\dot{B} = -D_1 E \quad \text{(Faraday)}, \qquad \dot{D} = D_1^\top H \quad \text{(Ampère)}, \tag{16}$$

which compactly reads

$$\dot{z} = J e = \underbrace{\begin{pmatrix} 0 & -D_1^\top \\ D_1 & 0 \end{pmatrix}}_{J} \underbrace{G_\theta^{-1}}_{\text{SPD}} z. \tag{17}$$

i.e., the special case of our update $\dot{z} = (J - R_\theta)G_\theta^{-1} z$ with $R_\theta \equiv 0$. Here $J$ is fixed entirely by signed incidences (mesh topology and orientation), while material/geometry enter only through the learned SPD metric blocks $G_\theta^{-1}$ (the constitutive law $\varepsilon^{-1}, \mu^{-1}$).

**Maxwell Structure as a Consequence.** Under the requirements used in the main text, the MeshFT thus recovers the source–free Maxwell update on arbitrary oriented meshes without explicitly postulating the Maxwell equations: orientation–aware incidences determine the conservative wiring $J$, and the *only* trainable physical freedom is the degree–wise SPD Hodge blocks in $G_\theta^{-1}$ (the material law). Unlike residual–based designs, we do not enforce a PDE loss. The structure rules out non-physical couplings by construction and focuses learning on the constitutive mapping.

**Note on Learning Freedom.** The structural constraint above *does not* fix any particular formula for the discrete Hodge stars. It only requires the degree-wise blocks of $G_\theta^{-1}$ to be SPD (and, if desired, local and permutation-equivariant). Thus the incidence wiring $J$ is fixed by topology, whereas all geometry/material dependence resides in $G_\theta^{-1}$—precisely the learnable degrees of freedom (e.g., metric structure and spatially varying $\varepsilon^{-1}, \mu^{-1}$) that can be identified from data without violating the topological structure.

### C.2. Degeneracy from Topology and Elimination of Spurious Modes

Because DEC encodes the chain–complex identity $D_{k+1}D_k = 0$, the interconnection $J$ above is *rank–deficient* (degenerate): it has a nontrivial kernel aligned with topological constraints. Two immediate invariants on closed, source–free domains are

$$D_2 \dot{B} = -D_2 D_1 E = 0, \tag{18}$$

$$D_0^\top \dot{D} = D_0^\top D_1^\top H = (D_1 D_0)^\top H = 0, \tag{19}$$

expressing discrete $\nabla \cdot B = 0$ and the charge–free $\nabla \cdot D = 0$, respectively. These invariants do not depend on the metric maps $G_\theta^{-1}$ and hold to machine precision, thereby suppressing non-physical (spurious) modes such as fake magnetic monopoles or artificial charge accumulation. Crucially, these invariants are not a post hoc regularizer but a direct consequence of the physical constraints structurally enforced by MeshFT. This topological guarantee is a major difference from methods that learn physical dynamics on meshes, which typically cannot preserve such invariants exactly.

**C.3. Relation to Symplectic–ODE (Nondegenerate) Formulations**

Symplectic ODE integrators preserve a global symplectic structure on a finite-dimensional phase space, e.g., in canonical coordinates $(q, p)$. However, this by itself does not encode mesh incidence identities such as $D_{k+1}D_k = 0$. As a result, Gauss-type constraints (e.g., discrete divergence constraints) are not automatically preserved unless the discretization is explicitly constraint-compatible, and spurious modes can appear over long rollouts.

By contrast, fixing the interconnection via $J$ builds the topology into the dynamics. $J$ is assembled from incidences, so the relations like $D_{k+1}D_k = 0$ are hard-wired, and the corresponding constraint subspaces are preserved mechanically. This cleanly separates topology/interconnection ($J$) from metric/constitutive content ($G$ and $R$), which can then carry geometry/material information.

# D. Details of Experiments and Additional Results

**Link to Theorem 3.3 (notation).** In this section, we use the standard FEM symbols $M$ (0-form Hodge on nodes) and $W$ (1-form Hodge on edges) as the concrete *metric blocks* of Theorem 3.3. Throughout the experiments given in this paper, we assumed $M$ and $W$ are *state-independent*. With canonical packaging $z = (q, p) \in C^k \oplus C^k$, we define the canonical momentum as $p := M\dot{q}$ (so that the kinetic energy is $\frac{1}{2}\dot{q}^\top M\dot{q} = \frac{1}{2}p^\top M^{-1}p$). Then

$$H(z) = \tfrac{1}{2}q^\top K q + \tfrac{1}{2}p^\top M^{-1}p, \qquad K = D_0^\top W D_0, \tag{20}$$

so

$$e = \nabla H(z) = \begin{pmatrix} Kq \\ M^{-1}p \end{pmatrix} = G^{-1}z, \qquad G^{-1} = \begin{pmatrix} K & 0 \\ 0 & M^{-1} \end{pmatrix}. \tag{21}$$

The conservative interconnection is the canonical symplectic matrix $J = \begin{pmatrix} 0 & I \\ -I & 0 \end{pmatrix}$, and the dynamics is $\dot{z} = (J - R(z))e$ with $R(z)^\top = R(z) \succeq 0$. Equivalently, under the mixed packaging $z = (x^{(k)}, x^{(k+1)}) \in C^k \oplus C^{k+1}$, $J = \begin{pmatrix} 0 & -D_0^\top \\ D_0 & 0 \end{pmatrix}$ and absorbing $D_0$ into $K = D_0^\top W D_0$ yields the canonical form above.

**D.1. Analytic Plane-Wave on Periodic 2D Meshes**

**Meshes and Geometry.** On a torus, we use (i) a periodic axis-aligned grid and (ii) a Delaunay triangulation. We report results on regular grid or Delaunay mesh at a nominal $32 \times 32$ resolution. Node dual areas $V_0$ come from cell/barycentric areas and edge weights $V_1^{-1}$ use nonnegative cotangent weights computed with periodic minimum–image distances, with small quantile floors to avoid degenerate areas/edges. Edge features are $(\Delta x, \Delta y, \|e\|)$ and node features are $(x, y, V_0)$.

**Data Generation.** Each training pair $(z_t, z_{t+\Delta t})$ uses the canonical state $z = (q, p)$ with a single traveling wave

$$q(t) = a\sin(k^\top x - \omega t + \phi), \qquad \omega = c\|k\|, \qquad p = V_0\dot{q}(t),$$

where $x$ are nodal coordinates on the periodic box $[0, L]^2$ and $V_0$ is the node volumes. For each sample we draw a wavenumber $k = \frac{2\pi}{L}(k_x, k_y)$ with $k_x = s_x U_x$, $U_x \sim \text{Unif}\{1, \cdots, 4\}$, $s_x \in \{\pm 1\}$ equiprobable, and $k_y = s_y U_y$, $U_y \sim \text{Unif}\{0, \cdots, 4\}$, $s_y \in \{\pm 1\}$. The zero mode $(k_x, k_y) = (0, 0)$ is excluded. Independently, we sample the phase $\phi \sim \text{Unif}[0, 2\pi)$, amplitude $a \sim \text{Unif}[0.5, 1.5]$, and start time $t_0 \sim \text{Unif}[0, 2\pi)$. We then set

$$z_t = (q(t_0), V_0\dot{q}(t_0)), \qquad z_{t+\Delta t} = (q(t_0 + \Delta t), V_0\dot{q}(t_0 + \Delta t)).$$

Randomness is controlled via synchronized seeds. Splits and batches are identical across models.

**Hodge Parametrization.** MeshFT-Net fixes the interconnection $J$ via the signed incidences $\{D_k\}$ and *learns only the metric*. A small geometry–conditioned Hodge maps node/edge features to positive stars:

$$M_i = V_{0,i}\,\sigma(\phi_{\text{node}}(x_i, y_i, V_{0,i})), \qquad W_e = V_{1,e}^{-1}\,\sigma(\phi_{\text{edge}}(\Delta x_e, \Delta y_e, |e|)),$$

with $\sigma = \text{softplus}$. Here, $\phi_{\text{node}}$ and $\phi_{\text{edge}}$ are *state–independent* MLPs (2 layers, width 64) that take standardized geometry as input and output a scalar log–scale, and applying $\sigma$ ensures $M_i > 0$ and $W_e > 0$. This geometry-conditioned Hodge is used throughout the experiments below, unless otherwise stated.

*Table 9.* One-step MSE and energy drift by models trained on sufficient amount of training data for the regular grid data. Lower is better and drift closer to 0 is better. All numbers are computed on held-out validation data. Mean over 3 seeds are reported.

| Model | One-step MSE | Energy drift |
|-------|--------------|--------------|
| MeshFT-Net | $\mathbf{1.4 \times 10^{-7}}$ | $\mathbf{7.2 \times 10^{-3}}$ |
| MGN | $1.2 \times 10^{-6}$ | 12.0 |
| MGN-HP | $5.7 \times 10^{-4}$ | 7.3 |
| HNN | $4.3 \times 10^{-7}$ | $6.3 \times 10^{-1}$ |

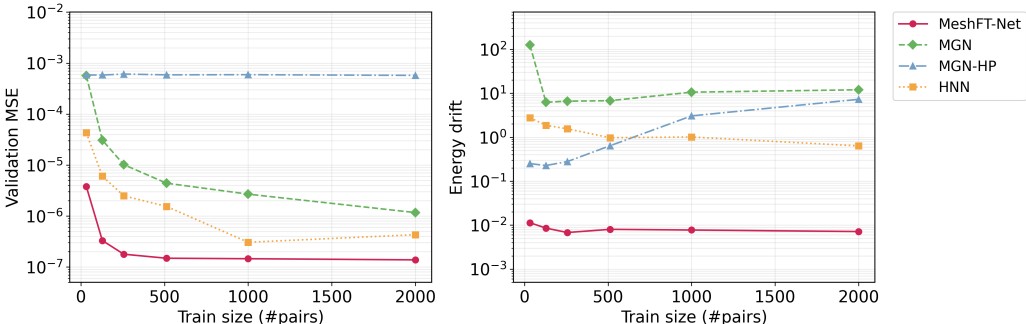

*Figure 5.* Relationship between the size of training dataset and one-step MSE (left) and rollout energy drift (right) for random delaunay mesh.

**Metrics and Hyperparameters.** For evaluation, a shared theory Hodge $(M, W) = (V_0, c^2 V_1^{-1})$ defines the physical norm used for relative error and for energy-drift over open-loop rollouts ($\Delta t$=0.002, $T$=200). Training runs for 10 epochs with a mini-batch size of 8 on 2000 training pairs and 256 validation pairs. The wave speed in the analytic solution is set to 1.0. MGN and MGN-HP use hidden width 64 with 4 message-passing layers. The network for predicting Hamiltonian in MGN-HP is configured identically (hidden 64, 4 layers). The HNN likewise uses hidden width 64 with 4 layers. The other exact hyperparameters are provided in the released code.

**Additional Results.** For completeness, in addition to the regular-grid results reported in the main text (Sec. 5.1), this appendix presents the same analytic plane-wave benchmark on periodic Delaunay triangulations under identical training protocols. The Delaunay results corroborate our main findings. MeshFT-Net maintains the lowest error and near-zero drift, while the baselines exhibit substantially larger drift with detailed numbers and visualizations provided in Table 9 and Fig. 5.

### D.2. Dissipative Benchmark

**Data Generation.** We benchmark *damped* plane-waves on a periodic 2D grid, using the canonical packing $x = (q, p)$. Samples are generated as

$$q(x, t) = A \, e^{-\gamma t} \sin(k^\top x - \omega t + \phi), \qquad \gamma \sim \text{Unif} \, [0.01, 0.1],$$

and we form one-step pairs $(x_t, x_{t+\Delta t})$ on a periodic $32 \times 32$ grid over $[0, 1)^2$, with $\Delta t = 0.002$, wave speed $c = 1.0$, and integer wavenumbers up to 6 (excluding the zero mode). The other parameters are same as the analytic plane-wave benchmark shown in Appendix D.1.

**Models and Integration.** Time stepping mirrors the conservative structure: MeshFT-Net uses KDK (with *exact half–step damping*, i.e., Strang split), HNN uses the same KDK-with-damping scheme, and MGN/MGN–HP are wrapped in a symmetric KDK integrator applied to their learned vector field $v(x)$. For MeshFT-Net and HNN we use nodewise Rayleigh dissipation: $\dot{p} = -Kq - Rp$ with rates $r_i \geq 0$ (equivalently $R = \text{diag}(r) \succeq 0$). Concretely, we use *GammaInferNet*—a small per-node damping MLP (message passing; width 64, 2 layers)—that ingests node/edge features, outputs raw rates $\tilde{r} \in \mathbb{R}$, and sets $r = \text{softplus}(\tilde{r}) \geq 0$. In continuous time the momentum channel obeys $\dot{p} = -Kq - Rp$, and in our Strang-split KDK step we apply the exact exponential decay over a step $\Delta t$: $p \leftarrow e^{-\frac{\Delta t}{2} R} p$  (before and after the conservative pass). MGN/MGN–HP *do not* receive an explicit damping operator and must infer dissipation from data.

**Metrics and Hyperparameters.** We report (i) one–step MSE and (ii) normalized energy drift over open-loop rollouts

*Table 10.* Details of physics metrics (computed on closed-loop rollouts with the shared *theory* Hodge $(M, W) = (V_0, c_{\text{speed}}^2 V_1^{-1})$). Frames are indexed $t = 0, \ldots, T$ (initial included).

| Diagnostic | Physical meaning and exact computation |
|---|---|
| Dispersion | Wave speed from the phase of $q$ at wavenumber $k$. Estimate $\hat{\omega}$ from weighted phase increments and set $\hat{c} = \hat{\omega}/\|k\|$. Reported as relative absolute errors vs. ground truth. |
| Canonical consistency | Check $p \approx M\dot{q}$ using central differences (sum over interior times $t = 1, \ldots, T-1$): $\dot{q}_t \approx \frac{q_{t+1} - q_{t-1}}{2\Delta t}$, $p_t^{\text{mid}} = \frac{1}{2}(p_{t+1} + p_{t-1})$. Reported $$\frac{\sum_{t=1}^{T-1} \| p_t^{\text{mid}} - M\dot{q}_t \|_2^2}{\sum_{t=1}^{T-1} \| M\dot{q}_t \|_2^2}.$$ |
| PDE residual | Discrete wave equation mismatch with $K = D_0^\top W D_0$ (again $t = 1, \ldots, T-1$): $\ddot{q}_t \approx \frac{q_{t+1} - 2q_t + q_{t-1}}{(\Delta t)^2}$, $r_t = M\ddot{q}_t + Kq_t$. Reported $$\frac{\sum_{t=1}^{T-1} \|r_t\|_2^2}{\sum_{t=1}^{T-1} \left( \|M\ddot{q}_t\|_2^2 + \|Kq_t\|_2^2 \right)}.$$ |
| Equipartition | Time-averaged kinetic/potential balance: $T_t = \frac{1}{2} p_t^\top M^{-1} p_t$, $U_t = \frac{1}{2} q_t^\top K q_t$ ($= \frac{1}{2}(Bq_t)^\top W(Bq_t)$), $\langle T \rangle = \frac{1}{T+1} \sum_{t=0}^{T} T_t$, $\langle U \rangle = \frac{1}{T+1} \sum_{t=0}^{T} U_t$, and reported $$\frac{|\langle T \rangle - \langle U \rangle|}{\langle T \rangle + \langle U \rangle}.$$ |
| Momentum | Normalized range of total momentum over time: $m_t = \sum_{n=1}^{N} p_t[n]$. Reported $$\frac{\max_{0 \leq t \leq T} m_t - \min_{0 \leq t \leq T} m_t}{\frac{1}{T+1} \sum_{t=0}^{T} \sum_{n=1}^{N} |p_t[n]|}.$$ |

($T = 200$) comparing with the true trajectory including the dissipation. Training runs for 20 epochs with a mini-batch size of 16 on 4000 training pairs and 256 validation pairs. MGN and MGN-HP use hidden width 64 with 4 message-passing layers. The network for predicting Hamiltonian in MGN-HP is configured identically (hidden 64, 4 layers). The HNN likewise uses hidden width 64 with 4 layers. The other exact hyperparameters are provided in the released code.

### D.3. Physically-Consistency Benchmark

**Meshes and Geometry.** We use periodic axis–aligned grids on the torus $[0, 1)^2$. The grid is $32 \times 32$, which is same as the analytic plane-wave benchmark shown in Appendix D.1.

**Data Generation.** Training pairs $(z_t, z_{t+\Delta t})$ come from analytic plane-waves $q(t) = a\sin(k^\top x - \omega t + \phi)$, with $\omega = c\|k\|$ and $c = 1.0$. Per sample we draw integer wavenumbers up to 6 (exclude the zero mode), a phase $\phi \sim \text{Unif}[0, 2\pi]$, amplitude $a \sim \text{Unif}[0.5, 1.5]$, and start time $t_0 \sim \text{Unif}[0, 2\pi]$.

**Metrics and Hyperparameters.** All physics diagnostics are computed with a shared *theory* Hodge, $(M, W) = (V_0, c^2 V_1^{-1})$, to ensure fair comparison, independent of a model's internal parameterization. The details of metrics for physical consistency are summarized in Table 10. Each hyperparameters are set as $\Delta t = 0.002$, $T = 200$, epochs $= 10$, batch $= 16$, train data size $= 4000$, and validation data size $= 256$. The other exact hyperparameters are provided in the released code.

In addition to physics diagnostics, we evaluate model–PDE agreement at the *vector-field* level and short-horizon behavior, using the same theory Hodge. (i) *Vector-field alignment*: at a batch of states $z$ we compute the PDE field $v_{\text{PDE}}(z) = [M^{-1}p, -Kq]$ and the model field $v_{\text{model}}(z)$, and report the cosine similarity and the relative $L_2$ error $\|v_{\text{model}} - v_{\text{PDE}}\|/\|v_{\text{PDE}}\|$. (ii) *Short-horizon rollout*: from the same initial states we roll out $T_{\text{short}}{=}16$ steps and report the final relative error. (iii) *Amplitude/phase recovery*: for the predicted field at $t = t_0 + T_{\text{short}}\Delta t$, fit $q(x) \approx a\sin(k{\cdot}x) + b\cos(k{\cdot}x)$ ($2{\times}2$ least squares) and report amplitude error $|\hat{A} - A|/|A|$ with $\hat{A} = \sqrt{a^2 + b^2}$, and phase error $|\hat{\phi} - \phi_{\text{eff}}|$ in degrees with $\phi_{\text{eff}} = \phi - \omega t$.

## D.4. OOD Generalization

**Data generation.** We use periodic 2D wave dynamics. Training and test pairs $(z_t, z_{t+\Delta t})$ are sampled from analytic plane-waves $q(x,t) = a\sin(k^\top x - \omega t + \phi)$ with $\omega = c\|k\|$. For each sample we draw integer wavenumbers up to 3 (excluding the zero mode). The other parameters are same as the analytic plane-wave benchmark shown in Appendix D.1. Training uses one–step teacher forcing. We generate three OOD settings by changing the test distribution relative to training: (i) **frequency** ($k_{\text{test}} > k_{\text{train}}$), (ii) **resolution** (coarse→fine grid), (iii) **wave speed** ($c_{\text{test}} \neq c_{\text{train}}$).

**Metrics and Hyperparameters.** All long-horizon errors and energy drift are evaluated with a theory Hamiltonian: on grids we use $(M, W) = (V_0, c^2 I_e)$. We report (i) one-step MSE on the test distribution and (ii) normalized energy drift over the trajectory.

Training is performed on a $32{\times}32$ periodic grid with time step $\Delta t_{\text{train}} = 0.004$, and wave speed $c_{\text{train}} = 1.0$, using $4000$ training pairs, batch size 16, and 10 epochs. Testing uses $\Delta t_{\text{test}} = 0.004$, $T = 200$, and 512 test pairs. Scenario-specific shifts are: (a) *frequency*—test_kmax = 6, (b) *resolution*—a finer $64{\times}64$ grid, and (c) *parameter*—wave speed $c_{\text{test}} = 1.4$. MGN and MGN-HP use hidden width 64 with 4 message-passing layers. The network for predicting Hamiltonian in MGN-HP is configured identically (hidden 64, 4 layers). The HNN likewise uses hidden width 64 with 4 layers. The other exact hyperparameters are provided in the released code.

**Additional Results.** The full results are provided in Table 11.

## D.5. Validation on Real Physics Field Data from *The Well*

**Data generation.** We evaluate the *acoustic_scattering_discontinuous* subset of *The Well* (2D acoustics with discontinuous media). Fields are placed on a Cartesian grid of $256 \times 256$ nodes over $[-1, 1] \times [-1, 1]$ with mixed boundary conditions such as *reflecting* in $x$ and *open* in $y$. From each short sequence we build one–step pairs $(z_t, z_{t+\Delta t})$ in canonical packing $z = (q, p)$, where $q$ is pressure and $p = M\dot{q}$ with node mass $M = V_0$ (cell area). The $p$ used for training is discretely calculated based on $q$. The dataset time step is $\Delta t = 2/101 \approx 0.0198$. To emulate open $y$-boundaries consistently across time stepping, we apply a light *sponge* after each update: $p \leftarrow p - \gamma_{\text{bias}}(y)\, p\, \Delta t$, with the bias ramped over the top/bottom $8\%$ of the domain.

**Hyperparameters.** Training follows the runner settings: 5 epochs, batch size 2, and 800 steps/epoch (validation every 50 steps). For stability, MeshFT-Net employs CFL-based *substepping* with target Courant number 0.5 on $256 \times 256$ nodes and $\Delta t {\approx} 0.0198$ this yields about 8 substeps per data step. The other exact hyperparameters are provided in the released code.

**Parameterization** We parameterize the discrete Hodge operators with compact, data-driven MLPs that are evaluated *locally* on nodes and edges.

For each node $i$, we predict a positive mass scale $\rho_i > 0$ with a small node-MLP fed by geometric/context features, including coordinates $(x_i, y_i)$ and cell area $V_{0,i}$. We then set

$$M_i = \rho_i\, V_{0,i}, \qquad \log\rho_i = \tanh\big(\phi_{\text{node}}(\cdot)\big),$$

where $\phi_{\text{node}}$ is a two-layer MLP (width 32).

For each edge $e = (i, j)$, we assemble geometric edge features $(\Delta x_e, \Delta y_e, |e|)$ and feed them to an edge-MLP. The edge-MLP outputs a positive scale $\sigma_e > 0$, and we define

$$W_e = \sigma_e\, V_{1,e}^{-1}, \qquad \log\sigma_e = \tanh\big(\phi_{\text{edge}}(\cdot)\big),$$

with $\phi_{\text{edge}}$ a two-layer MLP (width 32). Here $V_{1,e}$ denotes the (diagonal) discrete Hodge star on edges (primal 1-forms).

*Table 11.* OOD generalization across three scenarios. Lower is better. Bold indicates the best and underline the second best in each column. Mean $\pm$ s.d. over 3 seeds are reported.

| | Frequency extrapolation ($k_{\text{test}} > k_{\text{train}}$) | | |
|---|---|---|---|
| **Model** | **One-step MSE** | **TSMSE** | **Drift** |
| MeshFT-Net | $\mathbf{4.1 \times 10^{-6} \pm 2.6 \times 10^{-6}}$ | $\mathbf{1.8 \times 10^{-1} \pm 1.1 \times 10^{-1}}$ | $\mathbf{5.1 \times 10^{-3} \pm 1.2 \times 10^{-3}}$ |
| MGN | $\underline{2.3 \times 10^{-5} \pm 1.8 \times 10^{-5}}$ | $9.0 \pm 11.9$ | $>100$ |
| MGN-HP | $6.5 \times 10^{-3} \pm 4.0 \times 10^{-3}$ | $\underline{6.2 \times 10^{-1} \pm 3.0 \times 10^{-1}}$ | $\underline{1.6 \times 10^{-1} \pm 9.3 \times 10^{-2}}$ |
| PI-MGN | $1.0 \times 10^{-3} \pm 1.4 \times 10^{-3}$ | $>100$ | $>100$ |
| HNN | $4.8 \times 10^{-5} \pm 5.9 \times 10^{-5}$ | $1.5 \pm 1.4$ | $4.5 \times 10^{-1} \pm 5.3 \times 10^{-1}$ |
| FNO | $3.5 \times 10^{-4} \pm 7.0 \times 10^{-4}$ | $1.9 \pm 1.6$ | $2.7 \pm 2.2$ |
| GraphCON | $8.7 \times 10^{-5} \pm 7.7 \times 10^{-5}$ | $69.0 \pm 1.1 \times 10^{2}$ | $96.5 \pm 1.2 \times 10^{2}$ |

| | Wave speed shift ($c_{\text{test}} \neq c_{\text{train}}$) | | |
|---|---|---|---|
| **Model** | **One-step MSE** | **TSMSE** | **Drift** |
| MeshFT-Net | $5.8 \times 10^{-6} \pm 5.9 \times 10^{-6}$ | $\underline{7.1 \times 10^{-1} \pm 3.9 \times 10^{-1}}$ | $\mathbf{1.7 \times 10^{-1} \pm 8.0 \times 10^{-3}}$ |
| MGN | $\mathbf{3.1 \times 10^{-6} \pm 2.6 \times 10^{-6}}$ | $1.7 \pm 1.1$ | $24.5 \pm 25.2$ |
| MGN-HP | $3.2 \times 10^{-3} \pm 2.7 \times 10^{-3}$ | $\mathbf{5.9 \times 10^{-1} \pm 2.9 \times 10^{-1}}$ | $2.5 \times 10^{-1} \pm 2.0 \times 10^{-1}$ |
| PI-MGN | $2.6 \times 10^{-4} \pm 2.9 \times 10^{-4}$ | $>100$ | $>100$ |
| HNN | $8.7 \times 10^{-6} \pm 1.1 \times 10^{-5}$ | $8.0 \times 10^{-1} \pm 4.4 \times 10^{-1}$ | $\underline{2.4 \times 10^{-1} \pm 1.2 \times 10^{-1}}$ |
| FNO | $\underline{5.7 \times 10^{-6} \pm 4.5 \times 10^{-6}}$ | $9.6 \times 10^{-1} \pm 5.6 \times 10^{-1}$ | $1.6 \pm 2.0$ |
| GraphCON | $2.6 \times 10^{-4} \pm 2.9 \times 10^{-4}$ | $>100$ | $>100$ |

| | Resolution transfer ($32 \times 32 \rightarrow 64 \times 64$) | | |
|---|---|---|---|
| **Model** | **One-step MSE** | **TSMSE** | **Drift** |
| MeshFT-Net | $\mathbf{7.0 \times 10^{-7} \pm 5.4 \times 10^{-7}}$ | $\mathbf{5.1 \times 10^{-2} \pm 2.5 \times 10^{-2}}$ | $\mathbf{2.9 \times 10^{-3} \pm 9.2 \times 10^{-4}}$ |
| MGN | $8.6 \times 10^{-4} \pm 7.0 \times 10^{-4}$ | $5.3 \times 10^{-1} \pm 3.4 \times 10^{-1}$ | $3.4 \pm 2.7$ |
| MGN-HP | $1.5 \times 10^{-3} \pm 1.3 \times 10^{-3}$ | $5.8 \times 10^{-1} \pm 2.9 \times 10^{-1}$ | $5.6 \pm 4.4$ |
| PI-MGN | $8.6 \times 10^{-4} \pm 6.7 \times 10^{-4}$ | $>100$ | $>100$ |
| HNN | $8.6 \times 10^{-4} \pm 7.0 \times 10^{-4}$ | $3.6 \times 10^{-1} \pm 3.0 \times 10^{-1}$ | $2.0 \pm 4.7 \times 10^{-1}$ |
| FNO | $\underline{3.2 \times 10^{-5} \pm 3.7 \times 10^{-5}}$ | $>100$ | $>100$ |
| GraphCON | $8.9 \times 10^{-4} \pm 6.7 \times 10^{-4}$ | $\underline{3.2 \times 10^{-1} \pm 2.0 \times 10^{-1}}$ | $7.1 \times 10^{-1} \pm 1.3 \times 10^{-1}$ |

On a regular grid we set $V_{1,e} = 1$ (hence $V_{1,e}^{-1} = 1$). On irregular meshes $V_1$ is precomputed from geometry. All other hyperparameters follow the released code.

**Visualization.** From a validation sequence we form the canonical initial state $z_0 = [q_0, p_0]$. We then perform an open-loop $K$-step rollout with the symplectic KDK step, $\hat{z}^{j+1} = \text{MeshFT-Net}_{\Delta t}(\hat{z}^j)$ for $j = 0, \ldots, K - 1$, always feeding the *predicted* state into the next step. We set $K = 48$ and, in the main text, display five representative snapshots at frames $1, 12, 24, 36$, and $48$ (start/midpoints/end). At each step we record the predicted pressure $\hat{q}_j$ and render GT vs. MeshFT-Net contours side-by-side (using a color scale fixed by the GT frames).

## D.6. Nonlinear and Heterogeneous Benchmarks

This appendix provides the experimental details for the benchmarks summarized in Section 5.5. These benchmarks test MeshFT-Net outside the controlled linear plane wave setting. Unless otherwise stated, all reported values are means over 3 random seeds.

### D.6.1. Irregular Heterogeneous Deformable Body

**Objective.** This benchmark tests whether the topology–metric separation remains effective on a bounded irregular geometry with heterogeneous material response. Unlike the periodic wave benchmarks, the domain is non-periodic, contains an internal hole, uses an irregular Delaunay mesh, and has geometry-dependent constitutive coefficients. The purpose here is to evaluate the capability of MeshFT-Net in a more realistic deformable-body setting.

**Geometry and mesh.** We construct a two-dimensional body in the unit square with a circular hole centered at $(0.5, 0.5)$ with radius $0.18$. Interior points are sampled in the square and rejected if they fall inside the hole. We add points on the

*Table 12.* Scaling sweep on the irregular heterogeneous deformable-body benchmark. Runtime and memory are measured on a single NVIDIA H100 GPU. All values are means over 3 seeds. Lower is better except where noted.

| Nodes | TSMSE | Energy rel. MAE | Momentum variation | Infer. / step | Peak memory | Train batch |
|---|---|---|---|---|---|---|
| 464 | $5.85 \times 10^{-3}$ | $4.46 \times 10^{-2}$ | $2.75 \times 10^{-7}$ | 9.33 ms | 1.81 MB | 17.5 ms |
| 1232 | $3.60 \times 10^{-4}$ | $1.49 \times 10^{-2}$ | $1.87 \times 10^{-7}$ | 9.57 ms | 2.87 MB | 17.3 ms |
| 2256 | $1.95 \times 10^{-4}$ | $9.52 \times 10^{-3}$ | $1.68 \times 10^{-7}$ | 18.97 ms | 4.88 MB | 33.4 ms |

outer boundary and the hole boundary, triangulate the resulting point set by Delaunay triangulation, and remove simplices whose midpoint or edge midpoints cross the hole. The hole boundary is approximated by snapping detected boundary vertices to the prescribed circle and retriangulating. This yields a bounded, non-periodic, irregular mesh. It is a lightweight approximate meshing procedure rather than a high-order or quality-optimized boundary-fitted discretization. Node volumes are computed from simplex area sharing, and edge weights are computed from non-periodic cotangent weights with a small positive floor for numerical robustness.

**Heterogeneous constitutive field.** A spatially varying material field is assigned to nodes. In the two-dimensional setting, the field has the form

$$m(x, y) = 1 + 0.35 \sin(2\pi x) \cos(2\pi y) + 0.25 \exp\left(-\frac{(r - 0.22)^2}{2(0.05)^2}\right),$$

where $r$ is the distance from the hole center. Edge-level constitutive coefficients are obtained from the average material value at the two endpoints, together with local geometric factors depending on edge length and edge direction.

**Reference dynamics.** The state is $x = (q, p)$, where $q$ is the nodal displacement variable and $p$ is the corresponding momentum. Reference trajectories are generated by a nonlinear spring system on the irregular mesh. The conservative force is edge-based. Edge strains are computed from signed node–edge incidences, constitutive coefficients are applied on edges, and forces are accumulated back to nodes by the transpose incidence. We use no damping in this benchmark, so total momentum is a relevant diagnostic for the conservative action–reaction structure.

**Data generation and training.** Training trajectories are generated from localized mixed initial conditions. We use 32 training trajectories, 8 in-distribution validation trajectories, and 8 higher-amplitude OOD trajectories. Each trajectory has 256 time steps with $\Delta t = 0.0025$. Reference trajectories are generated with 8 internal substeps. Training uses one-step teacher forcing with batch size 8 for 10 epochs. The optimizer is AdamW with learning rate $10^{-3}$ and weight decay $10^{-6}$. The training amplitude range is $[0.08, 0.24]$, while the OOD amplitude range is $[0.28, 0.45]$.

**Models.** We compare MeshFT-Net with MGN, MGN-HP, HNN, and GraphCON on this benchmark. FNO is not used here because this benchmark is defined on a non-periodic irregular Delaunay mesh rather than a regular grid. MeshFT-Net fixes the same signed-incidence conservative wiring as in the main experiments and only learns metric factors from geometric and material features.

**Metrics.** We report TSMSE for held-out OOD rollouts, energy relative MAE, and the relevant structure diagnostic for each benchmark. Energy relative MAE compares the predicted and reference energy traces normalized by the initial reference energy, i.e., $\frac{1}{T+1} \sum_{t=0}^{T} |\widehat{E}_t - E_t|/(|E_0| + \varepsilon)$. Momentum variation is the normalized temporal range of total momentum, $(\max_t P_t - \min_t P_t)/(\frac{1}{T+1} \sum_t \sum_i |p_i(t)| + \varepsilon)$, where $P_t = \sum_i p_i(t)$. Results are shown in Table 6 (a).

**Additional results: scaling sweep.** We also evaluate the same heterogeneous deformable-body setting at three mesh resolutions. For this sweep, we reduce the number of training trajectories and epochs to keep the experiment lightweight: 16 training trajectories, 4 in-distribution trajectories, 4 OOD trajectories, and 6 training epochs. Runtime is measured on a single NVIDIA H100 GPU. Per-step inference latency is averaged after warm-up, and one-train-batch time measures a forward–loss–backward step without updating parameters. Peak memory is the CUDA peak memory recorded during the timing run. We also report a comparison of runtime, memory usage, and accuracy with baselines in Appendix D.10.

The scaling results given in Table 12 show that rollout accuracy improves from the coarsest mesh and remains low at larger resolutions, while runtime and memory grow moderately. Momentum variation stays at the $10^{-7}$ scale across all three meshes. This suggests that the observed momentum preservation is tied to the topology-fixed signed-incidence conservative wiring, not a consequence of a particular mesh resolution.

**Discussion.** The irregular mesh and internal hole test whether the signed-incidence conservative wiring remains effective

beyond periodic regular domains, while the spatially varying material field tests whether the learned metric-dependent factors can represent heterogeneous constitutive response. Thus, the results support the generality of MeshFT-Net: the same topology–metric separation achieves accurate long-horizon rollouts and preserves total momentum at the $10^{-7}$ scale. The same benchmark is used in Appendix D.8 to isolate the roles of the heterogeneous metric and the topology-fixed conservative wiring.

### D.6.2. ADDITIONAL NONLINEAR AND HETEROGENEOUS LATTICE BENCHMARKS

**Objective.** We further evaluate MeshFT-Net on several nonlinear lattice systems that go beyond the analytic plane-wave benchmark. The conservative interconnection remains state-independent and fixed by the signed incidence matrix. Nonlinearity enters through metric-dependent constitutive or dissipative factors.

**Common setup.** All systems use the canonical state $x = (q, p)$, where $q$ is the nodal field and $p = M\dot{q}$ is the conjugate momentum. Let $B$ be the signed node–edge incidence matrix and let $\xi = Bq$ denote oriented edge strain. The generic dynamics are

$$\dot{q} = M^{-1}p, \quad \dot{p} = -f(q) - \Gamma p, \quad f(q) = B^\top \tau(\xi) + \nabla_q U_{\text{site}}(q),$$

where $M$ is the nodal mass/Hodge block, $\tau(\xi)$ is the edge constitutive law, $U_{\text{site}}$ is an optional on-site potential, and $\Gamma \succeq 0$ is a Rayleigh damping operator when dissipation is present. Thus, the topology-fixed part is the incidence assembly through $B$ and $B^\top$, while the system-dependent terms are $\tau$, $U_{\text{site}}$, and $\Gamma$.

Unless otherwise stated, we use $32 \times 32$ grids, $\Delta t = 0.002$, 256 rollout steps, 32 training trajectories, 8 in-distribution validation trajectories, 8 OOD trajectories, and batch size 8, 10 training epochs. Reference trajectories are generated with 8 internal substeps. Training uses one-step teacher forcing and the same optimizer, loss weighting, and integrator choices as in the main experiments.

**Target systems.** We consider the following nonlinear systems.

*FPU-type nonlinear spring wave.* This benchmark introduces an edge constitutive nonlinearity:

$$f(q) = B^\top \tau(\xi), \qquad \tau_e(\xi_e) = W_e\left(k_2\xi_e + k_4\xi_e^3\right), \qquad \xi = Bq.$$

We use $k_2 = 1.0$ and $k_4 = 20.0$. This tests whether incidence-wired coupling remains effective when the edge stress is nonlinear in the oriented strain.

*Damped $\phi^4$ / Duffing–Klein–Gordon lattice.* This benchmark uses a nonlinear on-site potential and Rayleigh damping:

$$f(q) = B^\top(Wk_eBq) + M(\mu q + \lambda q^3), \qquad \Gamma = \gamma I.$$

Equivalently, this corresponds to the on-site potential $U_{\text{site}}(q) = \sum_i M_i\left(\frac{1}{2}\mu q_i^2 + \frac{1}{4}\lambda q_i^4\right)$. We use $k_e = 1.0$, $\mu = 1.0$, $\lambda = 10.0$, and $\gamma = 0.02$. This is the damped nonlinear lattice benchmark shown in Table 6 (b). It probes whether the model follows the true dissipative energy decay.

*State-dependent conservative coupling.* This benchmark keeps the signed-incidence sparsity fixed but lets the effective edge coupling vary with the current state:

$$f(q) = B^\top \tau(q, \xi), \qquad \tau_e(q, \xi) = W_e\, c_e(\bar{q}_e, \xi_e)\, \xi_e, \qquad \xi = Bq,$$

with $c_e(\bar{q}_e, \xi_e) = k_e\left(1 + \alpha_q\bar{q}_e^2 + \alpha_\xi\xi_e^2\right)$, $\bar{q}_e = \frac{q_i + q_j}{2}$ for $e = (i, j)$. We use $k_e = 1.0$, $\alpha_q = 1.0$, and $\alpha_\xi = 1.0$. This benchmark tests whether models remain accurate and physically stable when the interaction strength is no longer fixed, while the incidence wiring is unchanged.

*Sine-Gordon lattice.* This benchmark uses a periodic on-site potential:

$$f(q) = B^\top(Wk_eBq) + M\beta\sin q,$$

corresponding to $U_{\text{site}}(q) = \sum_i M_i\beta(1 - \cos q_i)$. We use $k_e = 1.0$ and $\beta = 4.0$. It tests a qualitatively different nonlinear on-site response from the polynomial $\phi^4$ potential.

**Models.** For all grid benchmarks, we compare MeshFT-Net with MGN, MGN-HP, HNN, FNO, and GraphCON. MeshFT-Net keeps the signed incidence wiring fixed and learn local metric-dependent factors from geometry and state features.

*Table 13.* Additional nonlinear benchmark results. Values are means over 3 seeds. Lower is better. Best values are shown in bold, and second-best values are underlined. Energy MAE denotes energy relative MAE. Mono. viol. and Freq. err denote the monotonicity-violation rate and the relative dominant-frequency error, respectively. Entries shown as $> 100$ exceeded the display cap or yielded non-finite rollout statistics. A "–" means that the auxiliary diagnostic is omitted because the primary rollout/energy statistics are non-finite or diverged.

**(a) FPU nonlinear spring wave**

| Model | TSMSE | Energy MAE | Mom. var. |
|---|---|---|---|
| MeshFT-Net | $\underline{3.04 \times 10^{-2}}$ | $\underline{1.71 \times 10^{-1}}$ | $\mathbf{4.75 \times 10^{-8}}$ |
| MGN | $> 100$ | $> 100$ | $> 100$ |
| MGN-HP | $> 100$ | $> 100$ | $> 100$ |
| HNN | $\mathbf{9.05 \times 10^{-3}}$ | $\mathbf{4.99 \times 10^{-2}}$ | $\underline{6.06 \times 10^{-2}}$ |
| FNO | $6.35 \times 10^{-2}$ | $3.95$ | $3.41 \times 10^{-1}$ |
| GraphCON | $> 100$ | $> 100$ | $> 100$ |

**(b) Damped $\phi^4$ lattice**

| Model | TSMSE | Energy MAE | Mono. viol. |
|---|---|---|---|
| MeshFT-Net | $\mathbf{2.54 \times 10^{-4}}$ | $\mathbf{8.70 \times 10^{-3}}$ | $\mathbf{3.25 \times 10^{-1}}$ |
| MGN | $> 100$ | $> 100$ | – |
| MGN-HP | $> 100$ | $> 100$ | – |
| HNN | $1.15 \times 10^{-2}$ | $3.01 \times 10^{-2}$ | $4.60 \times 10^{-1}$ |
| FNO | $1.70 \times 10^{-1}$ | $4.32 \times 10^{-1}$ | $6.38 \times 10^{-1}$ |
| GraphCON | $\underline{9.19 \times 10^{-4}}$ | $\underline{1.98 \times 10^{-2}}$ | $\underline{3.54 \times 10^{-1}}$ |

**(c) State-dependent conservative coupling**

| Model | TSMSE | Energy MAE | Mom. var. |
|---|---|---|---|
| MeshFT-Net | $\mathbf{1.62 \times 10^{-3}}$ | $\mathbf{5.87 \times 10^{-2}}$ | $\mathbf{5.85 \times 10^{-8}}$ |
| MGN | $> 100$ | $> 100$ | $> 100$ |
| MGN-HP | $> 100$ | $> 100$ | $> 100$ |
| HNN | $2.68 \times 10^{-1}$ | $89.6$ | $1.88$ |
| FNO | $5.89 \times 10^{-2}$ | $4.05 \times 10^{-1}$ | $\underline{2.18 \times 10^{-1}}$ |
| GraphCON | $\underline{1.15 \times 10^{-2}}$ | $\underline{1.26 \times 10^{-1}}$ | $3.46 \times 10^{-1}$ |

**(d) Sine-Gordon lattice**

| Model | TSMSE | Energy MAE | Freq. err |
|---|---|---|---|
| MeshFT-Net | $\mathbf{1.48 \times 10^{-6}}$ | $\mathbf{4.28 \times 10^{-4}}$ | $\mathbf{0.00}$ |
| MGN | $> 100$ | $> 100$ | – |
| MGN-HP | $> 100$ | $> 100$ | – |
| HNN | $1.24 \times 10^{-2}$ | $\underline{1.81 \times 10^{-2}}$ | $1.25 \times 10^{-1}$ |
| FNO | $1.83 \times 10^{-1}$ | $4.02 \times 10^{-1}$ | $3.54 \times 10^{-1}$ |
| GraphCON | $\underline{3.90 \times 10^{-3}}$ | $8.43 \times 10^{-2}$ | $\mathbf{0.00}$ |

**Metrics.** For the additional nonlinear benchmarks, we report OOD rollout TSMSE as the primary accuracy metric. We also report energy relative MAE, denoted as Energy MAE in Table 13, to measure agreement with the reference energy trace. Depending on the physical structure of each system, we include one additional diagnostic: momentum variation for conservative systems where total momentum should be preserved, monotonicity-violation rate for the damped $\phi^4$ system, and relative dominant-frequency error for the sine-Gordon system. Lower is better for all reported metrics.

**Results.** Table 13 summarizes the additional nonlinear benchmark results. Across the four nonlinear benchmarks, MeshFT-Net achieves the lowest OOD rollout TSMSE on three tasks and the second-lowest TSMSE on the FPU nonlinear spring benchmark, while retaining the best momentum diagnostic on the conservative FPU and state-dependent coupling tasks. The advantage is especially large on the sine-Gordon benchmark, where MeshFT-Net outperforms the strongest baseline other thatn MeshFT-Net by several orders of magnitude. On the state-dependent conservative-coupling benchmark, it also achieves substantially lower rollout error and energy error than all other baselines. The FPU benchmark shows that the signed-incidence model preserves the conservative action–reaction structure and remains competitive on nonlinear edge constitutive response, while the damped $\phi^4$ benchmark shows that the same topology-fixed interconnection, combined with learned metric-dependent and dissipative factors, can follow nonlinear dissipative dynamics.

Figure 6 also provides qualitative rollout snapshots for the damped $\phi^4$ benchmark in Table 13. The visualization compares MeshFT-Net with GraphCON, the strongest baseline on this benchmark other than MeshFT-Net, and shows that the lower rollout error and energy relative MAE correspond to visibly smaller spatial error over the rollout.

In addition, the physical diagnostics are consistent with the rollout errors. In the conservative FPU and state-dependent-coupling systems, MeshFT-Net keeps momentum variation orders of magnitude smaller than the strongest baselines other than MeshFT-Net, reflecting the preserved signed-incidence action–reaction structure. In the damped $\phi^4$ and sine-Gordon systems, MeshFT-Net gives the lowest energy relative MAE, and in the damped case it also gives the lowest monotonicity violation rate. These results support the theorem-derived modeling rule: fix the signed-incidence conservative interconnection and learn metric-dependent constitutive and dissipative factors.

### D.7. State-Dependent Conservative-Coupling Ablation

**Data generation.** We construct a fully controlled nonlinear system that follows the pointwise reduction form with a state-dependent conservative interconnection. The state is a three-channel field $z = (h, m_x, m_y)$ on a periodic $H \times W$ grid. We generate trajectories from the conservative dynamics

$$\dot{z} = J_{\text{true}}(z)\, e_{\text{true}}(z), \quad e_{\text{true}}(z) = G_{\text{true}}\, z,$$

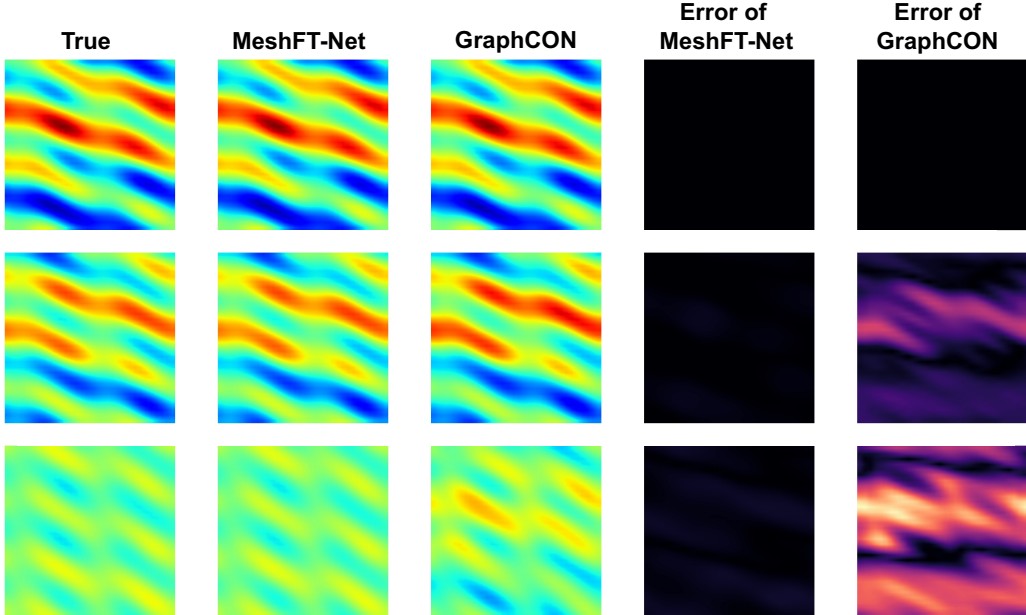

*Figure 6.* Qualitative rollout snapshots for the damped $\phi^4$ lattice benchmark. Rows show representative rollout times. Columns show the ground truth, MeshFT-Net, GraphCON, and the corresponding absolute error maps. GraphCON is the strongest baseline other than MeshFT-Net on this benchmark in Table 13. MeshFT-Net better preserves the spatial pattern over the rollout and yields smaller error maps, consistent with its lower TSMSE and lower energy relative MAE.

where $G_{\text{true}} = \text{diag}(g_0, 1, 1)$ is fixed and state-independent. The interconnection $J_{\text{true}}(z)$ keeps the signed-incidence skew wiring of the grid and is modulated only through orientation-even, positive per-edge gains. Concretely, we implement $J_{\text{true}}(z)$ as a conservative incidence-wired coupling that uses periodic cell-to-edge averaging and edge-to-cell divergence/adjoint-gradient operators. The gains are computed from orientation-even features $(h, m_x^2, m_y^2)$ to ensure invariance under momentum sign flips, and are normalized per sample by a mean-one gauge fix to remove scale non-identifiability. Trajectories are integrated by RK4 with time step $\Delta t$ for a fixed horizon and then converted into $k$-step supervision pairs $(z_t, z_{t+k})$ by unrolling $k$ steps of the learned one-step map.

**Models.** We evaluate a $2 \times 2$ ablation over the conservative coupling and metric-side parameterization. For the conservative interconnection, we compare a state-independent coupling against a state-dependent coupling that retains the same signed-incidence sparsity/sign pattern and predicts only the local gains from $(h, m_x^2, m_y^2)$. For the energy metric, we compare a diagonal per-cell SPD map against a full $3 \times 3$ SPD per-cell map parameterized via a Cholesky factor. In all cases, the update is advanced by the same explicit Heun stepper.

**Training.** All variants are trained on the same train/validation split by holding out entire trajectories to avoid temporal leakage. Training uses $k$-step supervision by unrolling the learned one-step map for $k$ steps and minimizing mean-squared error between $z_{t+k}$ and the reference $k$-step target. All hyperparameters (optimizer, learning rate, epochs, batch size, and gradient clipping) are shared across variants.

**Hyperparameters.** We use $H{=}W{=}64$, $\Delta t{=}0.02$, rollout length $T{=}600$ steps, and $k{=}8$ for $k$-step supervision. We generate 4 trajectories with trajectory length 260 and split them by trajectory IDs. Optimization uses Adam with learning rate $10^{-3}$, batch size 6, 5 epochs, and gradient clipping at 1.0, with no weight decay. All hyperparameters are shared across the four variants.

**Metrics.** We report TSMSE, defined as the time-average of the per-step mean squared error over the rollout horizon, and energy drift $|E(T) - E(0)|$ under the fixed reference energy induced by $G_{\text{true}}$.

$$E(z) = \tfrac{1}{2} \sum_{i,j} \Big( g_0\, h_{ij}^2 + m_{x,ij}^2 + m_{y,ij}^2 \Big), \qquad \text{Drift}_E = |E(T) - E(0)|,$$

where $T$ is the final rollout time. Reported numbers are means over 3 random seeds.

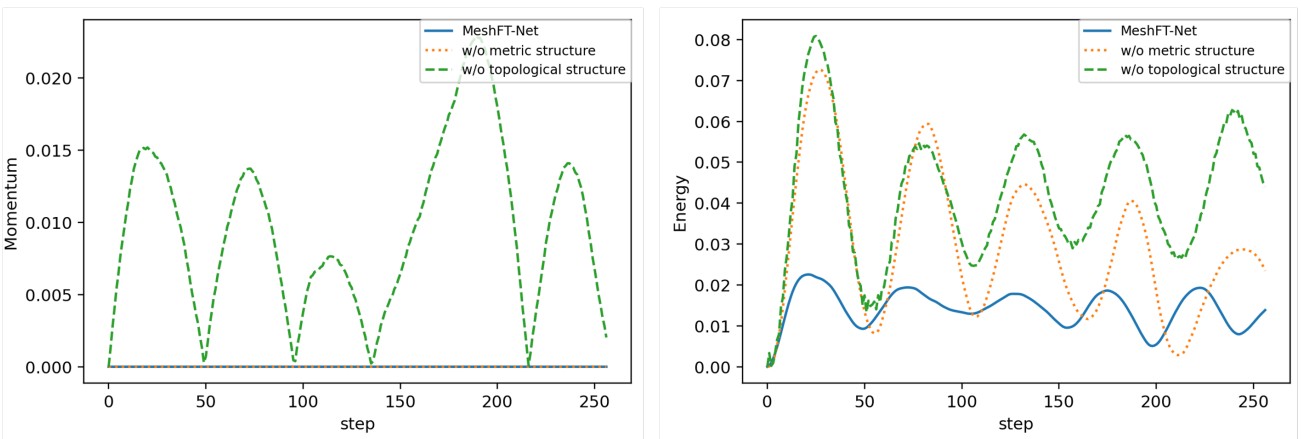

*Figure 7.* Topology–metric separation ablation on the heterogeneous-elasticity benchmark. Left: total momentum over a held-out rollout. Right: energy error over the same rollout. Restricting the heterogeneous metric mainly degrades energy fidelity while preserving near-constant momentum. In contrast, breaking the topology-fixed signed-incidence wiring causes visible momentum drift, consistent with the loss of the incidence-induced action–reaction structure.

### D.8. Topology–Metric Separation Ablations on Heterogeneous Elasticity

**Objective.** We provide implementation details for the topology–metric separation ablation reported in Section 5.6. Unlike the analytic ablations given in Appendix D.9, which selectively violate individual assumptions of Theorem 3.3 on a controlled wave benchmark, this experiment tests the topology–metric separation on the heterogeneous-elasticity benchmark: a bounded deformable body with a hole, an irregular non-periodic mesh, and geometry-dependent constitutive response. The goal is to isolate which empirical failures are caused by restricting the metric side and which are caused by breaking the topology-fixed conservative wiring.

**Common setup.** The state is $x = (q, p)$, where $q$ is the nodal displacement variable and $p$ is the corresponding momentum. The body is sampled in the unit square with a circular hole. Boundary and interior points are triangulated, and simplices crossing the hole are removed. A spatially varying material field is assigned to nodes and then lifted to edge-level constitutive coefficients using local edge geometry and node/edge material features. The reference trajectories are generated by a nonlinear spring system with geometry-dependent edge coefficients. All models are trained on the same trajectories, optimizer, supervision, and time step, and all reported numbers are means over 3 seeds.

**Variants.** Let $B$ denote the signed node–edge incidence matrix, so that $(Bq)_e = q_j - q_i$ for an oriented edge $e : i \to j$. The full MeshFT-Net uses the topology-fixed conservative force

$$f(q) = B^\top \tau(q), \qquad \tau_e(q) = w_e\, c_\theta(a_e)\, (Bq)_e,$$

where $w_e$ is a fixed geometric edge weight, $a_e$ denotes local geometry/material features, and $c_\theta(a_e) > 0$ is the learned heterogeneous metric/constitutive factor.

The w/o metric variant keeps the same signed-incidence wiring $B^\top (\cdot) B$, but replaces the learned edge-dependent coefficient $c_\theta(a_e)$ by a single learned positive scalar shared by all edges. Thus, this variant preserves the topology-fixed conservative interconnection but removes spatially heterogeneous metric capacity. The w/o topology-fixed variant keeps local learned metric factors, but replaces the pure incidence strain $(Bq)_e = q_j - q_i$ by a translation-breaking local strain $\widetilde{\xi}_{ij} = (q_j - q_i) + \frac{\varepsilon}{2}(q_i + q_j)$ with $\varepsilon = 0.05$. This variant remains local and keeps metric learning, but it is no longer a pure signed-incidence conservative wiring and does not preserve rigid translation symmetry. Therefore total momentum preservation is no longer guaranteed.

**Discussion.** Table 7 in the main text reports aggregate rollout diagnostics, and Figure 7 visualizes the corresponding momentum and energy-error time series. Together, they show that the ablation separates two failure modes. Restricting the heterogeneous metric increases TSMSE and energy relative MAE, but the momentum variation remains at the $10^{-7}$ scale because the signed-incidence wiring is unchanged. This indicates that the metric side is responsible for representing spatially varying constitutive scales and energy fidelity. In contrast, breaking the topology-fixed wiring causes a much larger momentum variation, even though local metric learning is retained. This occurs because the modified edge force no longer

*Table 14.* Ablation results. Each number is mean for 3 seeds.

| Model | One-step MSE | Energy drift | Energy injection | Momentum |
|---|---|---|---|---|
| MeshFT-Net | $\mathbf{4.142 \times 10^{-6}}$ | $\mathbf{2.03 \times 10^{-3}}$ | $\mathbf{8.75 \times 10^{-3}}$ | $4.85 \times 10^{-8}$ |
| No-Orientation $|D_k|$ | $1.08 \times 10^{-4}$ | 13.7 | $> 100$ | $> 100$ |
| Scrambled-Topology | $3.85 \times 10^{-5}$ | 13.1 | 31.1 | $4.90 \times 10^{-8}$ |
| Indefinite-Metric | $4.145 \times 10^{-6}$ | $2.30 \times 10^{-3}$ | $9.85 \times 10^{-3}$ | $\mathbf{4.28 \times 10^{-8}}$ |
| Learned-$J$ (PSD metric) | $4.37 \times 10^{-6}$ | $1.34 \times 10^{-1}$ | $4.59 \times 10^{-1}$ | $7.79 \times 10^{-3}$ |
| Learned-$J$ (free metric) | $4.16 \times 10^{-6}$ | 8.25 | 14.8 | $9.37 \times 10^{-2}$ |

*Table 15.* Mean inference and training latency (ms), peak training memory (MB), and TSMSE for four models evaluated on the analytic wave benchmark at three grid resolutions ($32 \times 32$, $64 \times 64$, $128 \times 128$). All values are averages over 3 random seeds.

| Model | Params | Infer [ms] | Train [ms] | Peak MB (train) | TSMSE |
|---|---|---|---|---|---|
| **32×32** | | | | | |
| MeshFT-Net | 642 | 1.74 | 3.32 | 18.7 | $7.8 \times 10^{-4}$ |
| MGN | 108,930 | 2.58 | 8.07 | 151.9 | $3.5 \times 10^{-1}$ |
| MGN-HP | 217,795 | 2.65 | 26.0 | 462.5 | $3.8 \times 10^{-1}$ |
| HNN | 217,602 | 17.1 | 50.6 | 976.7 | $3.4 \times 10^{-1}$ |
| **64×64** | | | | | |
| MeshFT-Net | 642 | 2.43 | 3.35 | 67.5 | $1.4 \times 10^{-4}$ |
| MGN | 108,930 | 5.48 | 14.4 | 596.5 | $3.0 \times 10^{-1}$ |
| MGN-HP | 217,795 | 5.47 | 45.5 | 1832 | $3.0 \times 10^{-1}$ |
| HNN | 217,602 | 29.9 | 94.4 | 3883 | $3.0 \times 10^{-1}$ |
| **128×128** | | | | | |
| MeshFT-Net | 642 | 2.70 | 3.82 | 262.8 | $6.5 \times 10^{-5}$ |
| MGN | 108,930 | 13.8 | 40.7 | 2379 | $1.2 \times 10^{-1}$ |
| MGN-HP | 217,795 | 13.8 | 142.5 | 7317 | $1.2 \times 10^{-1}$ |
| HNN | 217,602 | 104 | 306.9 | $1.55 \times 10^{4}$ | $1.2 \times 10^{-1}$ |

respects the action–reaction relation induced by the signed-incidence structure. Thus, this ablation clarifies the practical interpretation of the topology–metric separation.

**Relation to the following ablations.** The ablation here focuses on the practical topology–metric separation in a more realistic irregular geometry. The next subsection complements it with principle-level ablations on the analytic wave benchmark, where topology, orientation, metric positivity, and learned interconnection structure are selectively violated and evaluated under a common reference energy.

### D.9. Ablations on Topology, Orientation, and Metric Structure

**Objective.** Guided by Theorem 3.3, we test which structural assumptions matter in practice. Recall that locally $\frac{\partial F}{\partial z}(z) = (J - R(z))\,G(z)$ with $J^{\top} = -J$ assembled from the signed incidences $\{D_k\}$ (topology), $R(z)^{\top} = R(z) \succeq 0$ (dissipation), and $G(z) \succ 0$ (metric/constitutive). Our ablations selectively violate these ingredients while keeping data, optimizer, supervision, and the integrator fixed.

**Common setup.** All models are trained with one-step teacher forcing on analytic plane-wave pairs $x = [q, p]$ ($p = M\dot{q}$) on a $32 \times 32$ torus with step $\Delta t = 0.002$. Long-horizon evaluation uses $T = 200$ steps. For fair comparison, all rollout metrics use a *common* theory Hodge.

**Metrics.** (1) one-step MSE, (2) normalized energy drift $\frac{|E_t - E_0|}{|E_0|}$ over the rollout, (3) energy injection $E_{\mathrm{inj}} = \sum_{t=0}^{T-1} \max(E_{t+1} - E_t, 0)/(|E_0| + \varepsilon)$, and (4) momentum variation which is time variation of $\sum_n p_n$ on the torus, normalized by the mean $\ell_1$ amplitude, as in the physics-consistency test in Table 4. Lower is better for all.

**Compared Variants.** We consider the following variants for ablation study. The other mplementation details are provided in the released code. **MeshFT-Net (structured baseline).** $J$ is assembled from the signed incidences $\{D_k\}$ (topology +

orientation), the metric is positive ($G \succ 0$), and $R \equiv 0$ (conservative). *No assumptions are violated*. Energy and momentum are conserved up to integrator error, and dispersion is correct. **No-Orientation (orientation dropped).** Replace the signed incidence by an orientation–even map, violating (O). The resulting interconnection is no longer skew, $J^\top \neq -J$, so the conservative pass can inject/remove energy. We expect systematic energy drift and spurious momentum leakage. **Scrambled-Topology (topology broken).** Randomly re–pair the node–edge incidences so that $J$ is *not* the topology–assembled one in Theorem 3.3 and interface locality (L) is broken. Algebraic skew of $J$ is retained by construction, but under the common *theory* energy for evaluation, *expect* incorrect modal coupling and large long–horizon drift. **Indefinite-Metric (metric positivity broken).** Keep the signed topology (orientation preserved) but allow the edge–space weights to be signed, violating $G \succeq 0$. *Expect* nonphysical energy growth, positive energy injection, and unstable rollouts despite small one–step error. **Learned-$J$ (PSD metric).** Here $J$ is *learned* directly from data, not assembled from the signed incidences $\{D_k\}$ (topology identification dropped). It is constrained to remain *skew–symmetric* ($J^\top = -J$). The metric is kept positive ($G \succeq 0$). Thus, $J$ no longer reflects mesh topology even though it preserves the algebraic Hamiltonian symmetry. *Expect* stable yet biased dynamics under the *theory* energy: degraded dispersion/momentum behavior and moderate drift. **Learned-$J$ (free metric).** As above, $J$ is *learned* (not incidence–assembled) and remains *skew–symmetric* by construction, but we drop nonnegativity on the learned gains so the induced metric need not be PSD. This preserves the algebraic skew property of $J$ while violating metric positivity. *Expect* stronger bias, larger energy drift, and more energy injection than in the PSD case.

**Results.** Table 14 reports results of ablation study. Each number is mean for 3 seeds.

**Discussion.** *Topology (signed $\{D_k\}$) is critical.* Breaking incidence assembly (Scrambled-Topology) or dropping orientations (No-Orientation) yields large energy growth even when one-step MSE remains modest. No-Orientation, which violates (O), also shows pronounced momentum blow-up due to loss of action–reaction pairing, confirming that orientation is essential for a skew-symmetric $J$. In addition, Learned-$J$ (PSD metric) is markedly more stable than Learned-$J$ (free metric), yet both underperform the structured MeshFT-Net. Fixing $J$ via the mesh incidences $\{D_k\}$ is essential for stability and fidelity. *Violating metric positivity breaks energy balance.* Indefinite-Metric variants can achieve similar one-step MSE to MeshFT-Net yet inject more energy, underscoring that $J^\top = -J$ (from signed $\{D_k\}$) and a positive energy metric ($G \succeq 0$) must be enforced *together*.

**Takeaway.** These ablations empirically support Theorem 3.3: the conservative interconnection must be *assembled from the signed incidences $\{D_k\}$* (topology and orientation) and the metric/constitutive block must be *positive-definite*; otherwise long-horizon stability, conservation properties, and physical fidelity deteriorate—even when one-step errors are comparable.

### D.10. Computational Cost Analysis

To assess practical efficiency and accuracy, we report mean per-step inference/training latency, peak training CUDA memory, and TSMSE on the analytic wave benchmark (Sec. 5.1), measured on a single NVIDIA H100 under identical configs. These wall-clock numbers are intended as references due to implementation- and kernel-level differences across methods. As summarized in Table 15, across the three resolutions ($32^2$, $64^2$, $128^2$) MeshFT-Net shows a favorable time/memory/accuracy profile in this setup: it is typically the fastest for inference and training, and its memory footprint is markedly smaller. Latency growth with resolution is also modest for MeshFT-Net, whereas other models increase more noticeably. In these runs, MeshFT-Net achieved lower TSMSE (e.g., $6.5 \times 10^{-5}$ at $128^2$), and it uses substantially fewer parameters. HNN tended to be slower and more memory-intensive on this benchmark.

