# OpenReview forum: "Mesh Field Theory: Port–Hamiltonian Formulation of Mesh-Based Physics"
_ICML.cc/2026/Conference — ICML 2026 regular_

### Official Review · Reviewer_PtYd · 2026-02-13

**Soundness:** 4
**Presentation:** 4
**Significance:** 4
**Originality:** 4
**Overall Recommendation:** 6
**Confidence:** 5

**Summary:**

The authors present a framework for learning port-Hamiltonian dynamics on unstructured graphs using a parameterized discrete exterior calculus formalism. The work is well presented and complete; it is properly contextualized in the literature, clearly explained, and carefully benchmarked. I strongly recommend this be accepted.

**Compliance With Llm Reviewing Policy:**

Affirmed.

**Key Questions For Authors:**

Typo? "While this work starts from a PDEstyle template for network evolution, we start from minimal physical principles and derive a local
port–Hamiltonian reduction on."

A minor quibble about the "In contrast, we deduce a local port–Hamiltonian reduction from minimal principles and use it as an architectural inductive bias without assuming a single global form a priori.". To my understanding, Lee et al's paper also learns from "minimal principles" in the sense that they learn black box energy/entropy functionals without assuming any form beyond symmetry in brackets. There is also some more recent work from Gruber that the authors may have missed. This does not detract from the strength of this paper as the authors did a great job covering the literature.

**Limitations:**

yes

**Strengths And Weaknesses:**

The method is sound. There is a recent body of work starting from earlier Hamiltonian/Lagrangian networks and more recently dissipative bracket formalisms in whose context this paper represents state-of-the-art. I work in this area and can authoritatively state that the math all checks out and that this looks good to me, with benchmarks to other methods highlighting a non-trivial distinction in performance. There are careful ablation studies in the appendix, etc, and it is well written.

This is a significant problem - while lots of works have puttered around with physics-informed ideas using penalties, this represents an important contribution toward integrating physical structure by design (much like other recent structure preserving learning efforts) without assuming a priori knowledge of the governing model (for example, while PINNs penalizes a known physics loss, in this formalism the physics follow from a learned energy functional). Outside of the physics-oriented application community, these techniques could have value to the broader ML community (for example Bronstein's group has looked at incorporating related geometric dynamics into GNN design for conventional graph analytics tasks).

While the method draws on many salient pieces from the literature (other works on data-driven discrete exterior calculus, hamiltonian/port-hamiltonian/metriplectic/generic formalisms) this is a novel contribution that is clean and simple to understand. Some other works (e.g. Gruber+Lee+Trask) achieve similar capabilities but with a much more complicated setup - this manuscript is, in my opinion, easy to understand and will be accessible to the ICML community.

A suggestion for a minor weakness: The benchmarks are a little weak: a plane wave, where the method shows orders-of-magnitude superior performance to other techniques plays to the strengths of the method in a bit of an unfair way (there is a trivial energy functional that can be written down in terms of kinetic energy and a quadratic potential energy functional for the linear wave equation which I expect is what this method is picking up, explaining how it can achieve 1e-9 accuracy). The paper could be further strengthened by including a benchmark for a dissipative dynamical system which does not have a known port-hamiltonian structure - this would address the question of whether the technique can find a "nearby projection" onto dissipative dynamics. There is an interesting benchmark with nonlinear advection $\nabla \cdot (\alpha + \beta q) q)$ which explores this a little, but is pretty contrived. Why not just do incompressible navier-stokes, which has clearer impact, rather than make up a contrived PDE?

---

> ### Author Rebuttal · Authors · 2026-03-30
>
> Thank you very much for the strong support and for the careful reading. We are especially grateful for your expert assessment of the mathematical soundness, the empirical distinctions, and the overall clarity of the paper.
>
> > **The benchmarks are a little weak**
>
> We also appreciate your benchmark suggestion. We agree that the plane-wave setting is controlled and partly aligned with the strengths of the method. We chose it because it allows clean tests of physical consistency, including energy behavior, momentum conservation, and frequency accuracy. We also agree that broader evaluation would strengthen the paper. To address this point, in the rebuttal we added 5 extra nonlinear, dissipative, and heterogeneous benchmarks, including a deformable body with geometry-dependent heterogeneous elasticity. Across these added tasks, **MeshFT-Net ranks first on 4 of the 5 tasks and second on the remaining 1 by held-out rollout TSMSE**. The heterogeneous elasticity result is especially encouraging, since MeshFT-Net stays accurate on a more realistic deformable-body setting while keeping momentum variation near 1e-7. Full results will be included in the revised paper. **Please see representative results shown in the response to Z4BZ**.
>
> > **Why not just do incompressible Navier–Stokes**
>
> We agree that incompressible Navier–Stokes would have clearer practical impact. We did not choose it here because our goal was a clean diagnostic test of the topology–metric decomposition. This is an important future direction, and to broaden the empirical scope in the rebuttal, we added several additional nonlinear and dissipative benchmarks.
>
> ### **Answers to each question**
>
> Regarding the quoted typo, we could not locate this exact sentence in the current submission, so it may correspond to an earlier version of the manuscript. In any case, we will carefully re-check the related wording and make sure the revised version is clear and consistent.
>
> On the literature comparison, we agree that our wording should be more precise. Our intent was not to suggest that Lee et al. do not also start from physical principles. Rather, we would like to emphasize that in our case, we derive a local topology-driven port-Hamiltonian reduction directly from the stated physical principles, and then use that reduction as the architectural bias. We will revise the manuscript accordingly and also cite the more recent related work by Gruber.
>
> Thank you again for the very positive and expert assessment.

---

> > ### Author Rebuttal · Reviewer_PtYd · 2026-04-03
> >
> > Great work, thanks for the comments.

---

> > > ### Author Response · Authors · 2026-04-03
> > >
> > > Thank you very much for the positive assessment and for the careful reading. We sincerely appreciate your encouraging feedback and your thoughtful engagement with the paper and our responses.

---

### Official Review · Reviewer_H9RN · 2026-03-09

**Soundness:** 3
**Presentation:** 3
**Significance:** 1
**Originality:** 2
**Overall Recommendation:** 4
**Confidence:** 2

**Summary:**

## Summary

This paper proposes **Mesh Field Theory (MeshFT)**, a structure-preserving framework for learning mesh-based physical dynamics. The authors derive a theoretical reduction showing that under several physical assumptions (locality, permutation equivariance, orientation covariance, and energy balance), mesh-based dynamics admit a **local port–Hamiltonian factorization** where conservative coupling is determined by mesh topology and only metric-dependent operators need to be learned. Based on this insight, the paper introduces **MeshFT-Net**, which fixes the topology-induced incidence structure and learns metric and dissipation operators. Experiments compare the proposed approach with several neural simulators such as MGN and HNN on synthetic wave propagation tasks and an acoustic scattering dataset. The authors claim improved physical consistency, energy stability, and out-of-distribution robustness.

While the theoretical motivation is interesting, the empirical evaluation is limited and does not convincingly demonstrate the practical advantages of the proposed framework.

**Compliance With Llm Reviewing Policy:**

Affirmed.

**Final Justification:**

They address most of my concerns. As I am not deeply familiar with this specific sub-area, I will increase my score while lowering my confidence.

**Key Questions For Authors:**

1. Evaluation on more complex physical systems.
The current experiments mainly involve simple wave propagation tasks. Can the authors evaluate the method on more challenging systems such as nonlinear fluid dynamics, deformable solids, or multiphysics simulations?

2. Scalability to realistic simulation settings.
How does the proposed approach scale with larger meshes, more complex geometries, or higher-dimensional systems?

3. Comparison with stronger baselines.
Can the authors provide comparisons with modern operator-learning approaches (e.g., neural operators) or other recent structure-preserving neural simulators?

4. Generalization beyond analytic benchmarks.
Many experiments rely on analytic plane-wave dynamics. Can the authors demonstrate performance on more diverse datasets where the underlying dynamics are less structured or more difficult to model?

5. Role of the topology–metric decomposition in practice.
The theoretical reduction provides a clear conceptual separation between topology-driven coupling and metric-dependent operators. Can the authors provide additional empirical analysis demonstrating how this decomposition contributes to improved stability, accuracy, or generalization?

**Limitations:**

yes

**Strengths And Weaknesses:**

## Strengths

1. **Physically motivated formulation**
The paper provides a principled perspective on mesh-based physical simulation by separating topology-driven coupling from metric-dependent components. This interpretation is conceptually interesting and aligns with ideas from discrete exterior calculus and structure-preserving numerical methods.

2. **Structured inductive bias**
By fixing topology-induced interactions and learning only metric and dissipation operators, the proposed architecture incorporates physically meaningful constraints that may improve stability.

3. **Energy consistency analysis**
The paper includes diagnostic experiments evaluating energy drift, physical consistency, and OOD behavior, which are relevant for learned simulators.

## Weaknesses

1. Experimental tasks remain overly simplified.
Most experiments focus on analytic plane-wave benchmarks and relatively simple wave dynamics on regular meshes. These tasks are well controlled and may not fully reflect the challenges faced by real-world physical simulations, such as strongly nonlinear dynamics, complex boundary conditions, or heterogeneous materials.

2. Datasets are limited in diversity and realism.
The datasets used in the evaluation are relatively small and largely synthetic. Although the acoustic scattering dataset provides a step toward more realistic evaluation, it remains unclear whether the method scales to more complex simulation scenarios commonly studied in scientific computing.

3. Evaluation metrics are somewhat narrow.
The evaluation mainly reports prediction errors and energy drift metrics. While these are useful diagnostics, they may not fully capture the behavior of learned simulators in challenging long-horizon or highly nonlinear regimes. Additional evaluations on stability, robustness, or task-driven performance would strengthen the empirical evidence.

4. Baseline comparisons remain incomplete.
Although the paper compares with several neural simulation approaches (e.g., MGN and HNN), the evaluation does not include some strong recent baselines such as modern neural operator methods or other structure-preserving learned simulators. This makes it difficult to clearly assess the relative advantages of the proposed framework.

5. Limited evidence of practical impact.
While the theoretical framework is interesting, the experiments do not yet convincingly demonstrate that the proposed inductive bias leads to substantial improvements in realistic physical simulation settings or downstream applications.

---

> ### Author Rebuttal · Authors · 2026-03-30
>
> Thank you for the thoughtful feedback. We agree that broader and more realistic evaluation would strengthen the paper. In the rebuttal, we therefore added nonlinear, dissipative, state-dependent, and heterogeneous benchmarks, together with scaling and ablation studies.
>
> ### **Clarification of our empirical goal**
> The central contribution is the **reduction theorem**. Based on this result, we design MeshFT-Net so that the topology-driven conservative structure is fixed, while only the metric-dependent and dissipative parts are learned. In this sense, the **experiments are designed to assess the value of this theorem-derived inductive bias**. Our empirical goal is therefore to verify that this principled bias improves stability, physical fidelity, and OOD robustness in the studied regime. Thus, the controlled benchmarks were chosen to allow physical consistency to be tested cleanly through quantities such as energy behavior and momentum conservation.
>
> ### **Answers to each question**
> > **1. Evaluation on more complex physical systems**
>
> We added **5 extra benchmarks** covering irregular meshes, nonlinear and heterogeneous settings, including dissipative dynamics, state-dependent coupling, and geometry-dependent heterogeneous elasticity on a deformable body. By rollout TSMSE, **MeshFT-Net ranks first on 4 of the 5 tasks and second on the remaining 1**. A notable added result is a *heterogeneous deformable-body benchmark on an irregular mesh*, where MeshFT-Net achieves the best rollout TSMSE while keeping momentum variation near zero. Full results will be included in the revised paper. **Please see representative results in our response to Z4BZ**.
>
> For completeness, the current paper also includes a limited irregular-mesh evaluation in **Appendix C.1**. In addition, regarding evaluation metrics (**Weakness 3**), the paper reports physics-consistency analyses using metrics beyond prediction error and energy drift, as shown in **Table 3**.
>
> >**2. Scalability to realistic simulation settings**
>
> We refer to **Appendix C.8 and Table 10**, which report inference time, training time, peak memory, and accuracy. To further address scalability in a more realistic setting, we also ran a scaling sweep on *a deformable body with a hole and geometry-dependent heterogeneous elasticity on an irregular mesh*.
>
> |Nodes|TSMSE|Energy rel. MAE|Momentum var|Per-step inference|Peak memory|One train batch|
> |-|-:|-:|-:|-:|-:|-:|
> |464|5.85e-03|4.46e-02|2.75e-07|9.33 ms|1.81 MB|17.5 ms|
> |1232|3.60e-04|1.49e-02|1.87e-07|9.57 ms|2.87 MB|17.3 ms|
> |2256|1.95e-04|9.52e-03|1.68e-07|18.97 ms|4.88 MB|33.4 ms|
>
> Across this range, memory grows moderately, and total momentum variation stays near `1e-7`, indicating that the conservative structure is preserved well in practice. These results strengthen the evidence that the method scales to larger meshes and more complex geometries beyond the original analytic wave settings.
>
> > **3. Comparison with stronger baselines**
>
> We agree that comparison to more advanced methods would strengthen the paper. For completeness, we note that the current paper includes the **neural-operator baseline FNO**, as well as GraphCON and PI-MGN, in addition to MGN, MGN-HP, and HNN shown in **Tables 3 and 4**. We also note that HNN and MGN-HP serve as structure-preserving baselines, while PI-MGN serves as a physics-informed mesh-simulation baseline.
>
> > **4. Generalization beyond analytic benchmarks**
>
> We added nonlinear, state-dependent, and heterogeneous benchmarks. Please see the answer to question 1.
>
> > **5. Role of the topology–metric decomposition in practice**
>
> To make this separation explicit in practice, the rebuttal adds a targeted ablation on **geometry-dependent heterogeneous elasticity**. We compare MeshFT-Net with two controlled ablations. The **w/o metric structure ablation** keeps the conservative topology-driven coupling but replaces the learned heterogeneous metric with a spatially uniform one. The **w/ broken topological symmetry ablation** keeps metric learning but breaks the translation-symmetry structure of the conservative coupling, which directly affects momentum preservation.
>
> |Model|TSMSE|Energy rel. MAE|Momentum var|
> |-|-:|-:|-:|
> |MeshFT-Net|**4.83e-03**|**5.32e-02**|**2.67e-07**|
> |w/o metric structure|1.50e-02|9.99e-02|2.78e-07|
> |w/o topological structure|1.41e-02|6.78e-02|1.26e-02|
>
> Removing metric structure mainly hurts **energy fidelity**, while momentum preservation stays nearly unchanged. In contrast, breaking the topological symmetry mainly hurts **momentum preservation**, while the energy error degrades more moderately. This supports that the topology–metric decomposition is not only conceptual, but also practically important for stability, accuracy, and generalization.
>
> We hope these additions address your concerns and make the empirical validation materially broader. Thank you again for the constructive feedback.

---

> > ### Author Rebuttal · Reviewer_H9RN · 2026-04-03
> >
> > Thank you for the detailed response. While it addresses several of my concerns and provides useful clarifications, some issues remain. Could you include visual results for the additional experiments? I would like to see qualitative evidence that supports your claims, rather than relying solely on numerical results or theoretical analysis.

---

> > > ### Author Response · Authors · 2026-04-03
> > >
> > > Thank you for this suggestion, and we sincerely appreciate your careful review. To adress this point, we have prepared qualitative visual results, which can be viewed here:
> > >
> > > https://anonymous.4open.science/api/repo/fig-link-2026-DBDE/file/figures_clean.pdf?v=f74456cc
> > >
> > > ___
> > >
> > >
> > > The file contains five figures. *Figures 1 and 2* show rollout snapshots for two added benchmarks: **the damped nonlinear lattice wave system** and **the deformable body with a hole and geometry-dependent heterogeneous elasticity**. In each case, we show the ground truth, MeshFT-Net, the strongest non-MeshFT baseline on that benchmark, and the corresponding error maps, so that the long-horizon qualitative differences can be seen directly. *Figure 3* shows the geometry and spatially varying material field for the irregular deformable-body benchmark, to clarify what physical setting is being modeled. Also, *Figure 4* shows the total momentum on the same held-out trajectory, illustrating that MeshFT-Net stays nearly indistinguishable from the ground truth while the baseline exhibits substantial drift.
> > >
> > > *Figure 5* shows ablation curves for the deformable-body benchmark. It is meant to clarify the practical role of the topology--metric decomposition. Removing metric structure mainly worsens energy accuracy, while removing topological structure destroys momentum preservation and also severely degrades long-horizon energy stability. These results will included in the revised paper.
> > >
> > > ___
> > >
> > > We hope these visual results provide a useful qualitative complement to the rebuttal tables and help make the essential contribution of MeshFT more transparent.

---

### Official Review · Reviewer_Z4BZ · 2026-03-11

**Soundness:** 3
**Presentation:** 3
**Significance:** 3
**Originality:** 3
**Overall Recommendation:** 5
**Confidence:** 4

**Summary:**

This paper proposes Mesh Field Theory and MeshFT-Net, which fix the topological conservative structure from the mesh and only learn the metric-dependent and dissipative parts, to improve stability, energy behavior, and physical fidelity in mesh-based simulation.

**Compliance With Llm Reviewing Policy:**

Affirmed.

**Final Justification:**

This paper has solid content, with theoretical proof and a variety of datasets to evaluate the model.

**Key Questions For Authors:**

see above

**Limitations:**

yes

**Strengths And Weaknesses:**

Strengths
1. The paper is built on a clear and principled foundation, from four minimal physics principles (locality, permutation equivariance, orientation covariance, and energy balance/passivity) and uses them to motivate and ground the MeshFT-Net.
2. Paper reported experiments are designed with checking TSMSE, energy drift, PDE residual, canonical consistency, momentum, OOD rollout, and data efficiency, which makes the evaluation convincing and aligned with the paper’s goals.
3. MeshFT-Net shows clear and strong advantages, especially on long-horizon rollout (TMSE), energy behavior, and physical consistency, and experiment results support the paper’s main claims quite well.

Weaknesses
1. The paper would be stronger with more experiments on broader benchmark datasets and more recent models.
2. Since the proposed method introduces additional structure and constraints, it would be helpful to report training/inference time and memory usage (computational cost).

---

> ### Author Rebuttal · Authors · 2026-03-30
>
> Thank you for the positive assessment of the motivation, evaluation, and empirical results. We are glad that the main goals of the paper came across clearly.
>
> (**Weakness 1**)
> We agree that broader benchmarks would further strengthen the paper. To further extend the empirical scope in the rebuttal, we added **5 extra benchmarks** covering irregular meshes, nonlinear conservative dynamics, dissipative nonlinear dynamics, state-dependent coupling, and geometry-dependent elasticity. In this added suite, we compare MeshFT-Net against MGN, MGN-HP, HNN, FNO where applicable, and GraphCON. Using held-out rollout TSMSE as the main metric, **MeshFT-Net ranks first on 4 of the 5 tasks and second on the other 1**. In addition, across these tasks we observe broadly improved invariant- or stability-related diagnostics, such as lower energy error or lower momentum variation, as illustrated by the representative results below. Overall, these added results broaden the empirical scope beyond the original benchmarks and further support the robustness and physical fidelity of MeshFT-Net across regimes. Representative added results are shown below.
>
> (**Weakness 2**)
> We refer to **Appendix C.8** and **Table 10**, where we report inference time, training time, peak memory, and accuracy under the same hardware and training setting. We will make this reference more explicit in the main text and bring these results forward in the revision.
>
> To further address scalability in a more realistic setting, we ran a scaling sweep on an **deformable body with a hole and geometry-dependent heterogeneous elasticity　on irregular mesh**. The table below reports held-out TSMSE, energy relative MAE, momentum variation, per-step inference latency, peak memory, and one-train-batch time on an Nvidia H100 GPU, averaged over **3 seeds**.
>
> |Nodes|TSMSE|Energy rel. MAE|Momentum var|Per-step inference|Peak memory|One train batch|
> |-|-:|-:|-:|-:|-:|-:|
> |464|5.85e-03|4.46e-02|2.75e-07|9.33 ms|1.81 MB|17.5 ms|
> |1232|3.60e-04|1.49e-02|1.87e-07|9.57 ms|2.87 MB|17.3 ms|
> |2256|1.95e-04|9.52e-03|1.68e-07|18.97 ms|4.88 MB|33.4 ms|
>
> These results do not yet establish truly large-scale deployment, but they do show that MeshFT-Net remains stable as the mesh size increases on a more realistic bounded *irregular* geometry. The rollout error improves substantially from the coarsest mesh and remains low at larger resolutions, while inference time, batch training cost, and memory grow moderately. At the same time, the momentum variation stays near `1e-7` across all three resolutions, which is consistent with the fact that **momentum preservation comes from the topology-driven conservative wiring**, rather than from the mesh resolution itself.
>
> ---
>
> We hope these additional results and clarifications address your concerns. Thank you again for the constructive feedback.
>
> ---
>
> ### **Representative added benchmark results**
> For brevity, we report only representative added results below. All values are means over 3 seeds. For readability, values larger than `1e2` are shown as `>100`. Full results for all added benchmarks will be included in the revised paper, and we can share the remaining results during the rebuttal period if helpful.
>
> **Irregular deformable body with geometry-dependent heterogeneous elasticity**
> This benchmark tests a bounded deformable body with a hole, an irregular mesh, and a heterogeneous geometry-dependent elastic response. Even in this more realistic setting, MeshFT-Net remains substantially more accurate than the learned baselines, while preserving total momentum almost exactly.
>
> |Model|　TSMSE|Momentum var|
> |-|-:|-:|
> |MeshFT-Net|**4.91e-03**|**3.09e-07**|
> |HNN|2.95e-02|1.85e-01|
> |MGN-HP|6.88e-02|1.11e+00|
> |MGN|2.01e-01|1.51e+00|
> |GraphCON|>100|>100|
>
> **Nonlinear state-dependent conservative coupling**
> This benchmark tests a conservative lattice system in which the effective coupling changes with the state. It is designed to probe whether the model remains accurate and physically stable when the interaction strength is no longer fixed. The very low momentum variation shows that MeshFT-Net preserves the conservative structure much better in this regime.
>
> |Model|TSMSE|Energy rel. MAE|Momentum var|
> |-|-|-|-|
> |MeshFT-Net|**1.623e-03**|**5.872e-02**|**5.854e-08**|
> |GraphCON|1.153e-02|1.258e-01|3.462e-01|
> |FNO|5.885e-02|4.047e-01|2.176e-01|
> |HNN|2.676e-01|8.961e+01|1.877e+00|
> |MGN-HP|>100|>100|>100|
> |MGN|>100|>100|>100|
>
> **Damped nonlinear lattice oscillator system**
> This benchmark tests dissipative nonlinear wave dynamics. It is a standard setting for studying relaxation toward lower-energy states in a nonlinear medium. Lower energy error here means that the model follows the true dissipation and decay of the system more faithfully over rollout.
>
> |Model|TSMSE|Energy rel. MAE|
> |-|-|-|
> |MeshFT-Net|**2.538e-04**|**8.699e-03**|
> |GraphCON|9.188e-04|1.978e-02|
> |HNN|1.145e-02|3.011e-02|
> |FNO|1.695e-01|4.317e-01|
> |MGN-HP|>100|>100|
> |MGN|>100|>100|

---

> > ### Author Rebuttal · Reviewer_Z4BZ · 2026-04-01
> >
> > The authors have put in the effort to test their model on more varied datasets. Although not getting a new baseline, which I wish to see like Transolver [1], a different architecture compared to the existing presented models.
> >
> > But this is trivial, maybe the authors could consider including it in the revised manuscript.
> >
> > [1] Wu, Haixu, Huakun Luo, Haowen Wang, Jianmin Wang, and Mingsheng Long. "Transolver: A fast transformer solver for pdes on general geometries." Proceedings of the 41 st International Conference on Machine Learning 2024.

---

> > > ### Author Response · Authors · 2026-04-01
> > >
> > > Thank you for your careful re-evaluation. We sincerely appreciate your raising both the score and the confidence, and we are grateful for your positive recognition of our effort to evaluate the method on more diverse datasets.
> > >
> > > We also agree that comparison with a different architectural baseline such as Transolver would further strengthen the paper. Thank you for this helpful suggestion. We will consider including it in the revised manuscript.

---

### Official Review · Reviewer_6a6T · 2026-03-12

**Soundness:** 3
**Presentation:** 2
**Significance:** 3
**Originality:** 3
**Overall Recommendation:** 5
**Confidence:** 3

**Summary:**

The paper describes in a formal language the requirements that a model for mesh based on discrete exterior calculus has to fulfill.

The paper proceeds to derive the main result about the reduction to a port-Hamiltonian of the dynamics on the mesh.

By arguing on adopting a simplified model, the paper introduces the learnable form where only two functions F and G shall be learned.

The paper experiments on analythic 2d plan-wave data, comparing with MeshGraphNets and its proposed extensions (MGN, MGN-HP, PI-MGN, HNN, FNO, and GraphCON).

The proposed model shows better consistency.

**Compliance With Llm Reviewing Policy:**

Affirmed.

**Ethical Review Concerns:**

no ethical consideration

**Final Justification:**

The authors will provide additional information on the DEC that would help understand the contribution of the paper.

**Key Questions For Authors:**

# Q1
Could you please help to understand better the meaning of the symbols used in the paper? Also about the concrete representation of the Hamiltonian operator in the complex?

The paper would benefit by a more accessible presentation,

# Q2
The experiments are focused on the one-step prediction. Is it possible to train the energy differently? you mentioned for example, using the residuals of the PDE.

**Limitations:**

There is no explicit section, but the paper is complete and the experiments show the capabilities and limitations of the approach. The authors also express in the text the limitation of their approach. "This assumption still covers many standard settings including linear waves, linear elastodynamics, and linear electromagnetics on fixed media, with heterogeneity entering through the metric and dissipation terms."
No negative societal impact is perceived from the paper.

**Strengths And Weaknesses:**

# soundness
The paper has a solid theoretical foundation and provides the mathematical formulation that is then used to model the learnable function.


# presentation
While formal, the paper assumes knowledge in the reader of the properties and formalism of DEC. It would be better if the authors provide at least in the annex a more intuitive and detailed presentation of DEC and the meaning of the operator Dk, and the effect on the dynamic equation. For example, I am not sure I understand when J is built from Dk and Dk^T Is Dk an operator or a matrix? is ^T a transpose operator, or is it the inverse of the boundary operator?

In general, the paper could benefit from a more understandable and accessible presentation of meaning of the mathematical tools ()

# significance
The paper advances the understanding and models for modelling physical equations over a mesh structure with applications in many domains.


# originality
The paper extends previous work on mesh-based operator learning.

---

> ### Author Rebuttal · Authors · 2026-03-30
>
> Thank you for the careful reading and for the positive assessment of the theory. We understand your main concern as clarity and accessibility of the presentation. In the revision, we will add a **notation table** in the appendix and make the explanation of the mathematical tools (DEC) more direct and intuitive, including a *concrete example of the coboundary matrix for simple mesh.*
>
> ### **Answers to each question**
> > **Q1 Could you please help to understand better the meaning of the symbols used in the paper?**
>
> Thank you for pointing this out. In our paper, $D_k$ is the discrete exterior derivative, also called the coboundary operator. After fixing an oriented mesh, it is represented by a sparse signed incidence **matrix**. Also, $D_k^{\top}$ is the **transpose**, not an inverse. For example, when $k=0$, it maps node values to oriented edge differences such as $q_j - q_i$ along an edge $i \to j$, which can be understood as the finite-difference counterpart of $\nabla q$. In this sense, the coboundary operator plays the role of a discrete grad, curl, or div, depending on the degree.
>
> In our reduction, $J$ is *always* built from these coboundary blocks. This is an important consequence of the reduction. In the state-independent case, we can write $J = [[0, -D_k^T], [D_k, 0]]$, so the conservative coupling is automatically fixed by topology and orientation and does not need to be learned. A concrete example from electromagnetic dynamics appears in Appendix B. There, one can clearly see the correspondence between continuous differential operators such as curl and the discrete coboundary operator.
>
> To make these ideas more accessible to broader readers, we will add a more intuitive explanation of the coboundary operator, together with a small explicit matrix example of $D_k$.
>
> > **Also about the concrete representation of the Hamiltonian operator in the complex?**
>
> Thank you for this question. Strictly, the Hamiltonian is the scalar function $H$. In the simplest quadratic case, $H(z) = \frac{1}{2} z^T G z$, so $e = G z$ and $\dot z = (J - R) G z$. At the same time, in MeshFT-Net, we learn $G$ and $R$ explicitly, while $H$ is introduced mainly as the theoretical object used to derive the reduction and the co-energy relation. In that sense, $H$ itself is not given a separate explicit representation beyond the induced metric form. We will revise the wording to make this point clearer and avoid confusion.
>
> > **Q2 The experiments are focused on the one-step prediction. Is it possible to train the energy differently? you mentioned for example, using the residuals of the PDE.**
>
> Yes, other training choices are possible. We use a one-step supervised loss because it is the simplest setup for isolating the effect of the structure bias. However, our main evaluation is based on long-horizon rollout, physical consistency, and OOD behavior, not only one-step error. Also, as noted in **Section 4**, the update can also be stacked and supervised after several substeps.
>
> It is also **possible in principle to use PDE residuals**. We did not use PDE-residual training here because one goal of MeshFT is to work without assuming an explicit PDE form or residual supervision. We will clarify this point more clearly in the revision.
>
> We hope these additions address your concerns with respect to presentation. Thank you again for the constructive feedback.

---

> > ### Author Rebuttal · Reviewer_6a6T · 2026-04-03
> >
> > I thank the authors for the calrifications. I am still not 100% sure I understand well what is happening.
> >
> > In the revisioned document, please also add the dimensions of the operators, for example if D_k is a sparse matrix, it will belong to a specific space $\mathbb R^{m \times m}$. I would be nice to undersand if this does not overload too much the presentation, all the object spaces.
> >
> > #Update
> > Thank you for the clarification and table. I am adding this here since it is not possible to give futher comments.
> >
> > I still find confusing the description of space $C^k$, could you add examples where 1) only nodes, 2) nodes and edges, 3) nodes, edges, and faces are used? these are probably the most common set up, while I suppose in general we can have d-dimensional hyper-triangles (not sure how to name the general object).

---

> > > ### Author Response · Authors · 2026-04-04
> > >
> > > Thank you again for this helpful suggestion. We agree that explicitly stating the operator dimensions and object spaces would make the notation much easier to follow. In the revised manuscript, **we will therefore add a compact notation table summarizing the main spaces and operators** at the beginning of *Section 3*:
> > >
> > > ___
> > >
> > >
> > > | Symbol | Meaning | Space / dimension |
> > > |---|---|---|
> > > | $n_k$ | number of $k$-cells, e.g. vertices, edges, or faces | scalar |
> > > | $C^k$ | space of discrete $k$-cochains, i.e. quantities attached to oriented $k$-cells (e.g. node values such as displacement, and edge values such as flux) | $C^k \cong \mathbb{R}^{n_k}$ |
> > > | $z$ | state variable | $z \in C^k \oplus C^{k+1}$ |
> > > | $H$ | Hamiltonian / energy function (scalar) | $H : C^k \oplus C^{k+1} \to \mathbb{R}_{\ge 0}$ |
> > > | $D_k$ | discrete exterior derivative (coboundary) | $D_k : C^k \to C^{k+1}$, $D_k \in \mathbb{R}^{n_{k+1}\times n_k}$ |
> > > | $D_k^\top$ | transpose of $D_k$ | $D_k^\top : C^{k+1} \to C^k$, $D_k^\top \in \mathbb{R}^{n_k\times n_{k+1}}$ |
> > > | $J$ | conservative interconnection operator | assembled from blocks of $D_k$ and $D_k^\top$ |
> > >
> > > ___
> > >
> > > We will also add a small explicit example on a **single oriented triangle**, together with a simple schematic figure. For reference, we also provide the following anonymous URL with a small schematic figure of the single oriented triangle example and the corresponding matrices $D_0$ and $D_1$:
> > >
> > > https://anonymous.4open.science/api/repo/fig-link-2026-2-8368/file/orientation_clean.pdf?v=b61a933e
> > >
> > >  Since this mesh has *3 vertices, 3 oriented edges, and 1 oriented face*, we have $n_0=3$, $n_1=3$, and $n_2=1$. For a chosen orientation, the boundary operators can be written as
> > > $$
> > > D_0 =
> > > \begin{bmatrix}
> > > -1 & 1 & 0\\\\
> > > 0 & -1 & 1\\\\
> > > 1 & 0 & -1
> > > \end{bmatrix},
> > > \qquad
> > > D_1 =
> > > \begin{bmatrix}
> > > 1 & 1 & 1
> > > \end{bmatrix}.
> > > $$
> > > We will also clarify that the signs in $D_0$ are determined entirely by the chosen edge orientation. *The orientation itself can be chosen arbitrarily*, but once it is fixed, the incidence matrix is determined uniquely. For each oriented edge, the head contributes $+1$, the tail contributes $-1$, and all other entries are $0$, or vice versa. Thus, these matrices contain only entries in $\{0,\pm1\}$, i.e. they are sparse signed incidence matrices fixed by mesh topology together with the chosen orientation. Although the single-triangle example appears dense, in an actual mesh $D_k$ is sparse, since only incident pairs of cells contribute nonzero entries.
> > > ___
> > >
> > > To avoid overloading the main presentation, we plan to keep the notation table in the main text and place the explicit triangle example in the appendix. We hope this will make the meaning of the operators and their spaces much easier to follow.

---

### Decision · Program_Chairs · 2026-04-30

**Decision:**

Accept (regular)

**Comment:**

The manuscript proposed a structure preserving framework for machine learning based physics simulation. The reviewers unanimously recommend the paper, and the authors have addressed concerns raised during the discussion process. After reading the manuscript, the meta-reviewer agrees that the paper is a solid contribution and should be accepted.